# Provable Multi-Region Affinity Enforcement and Constraint Satisfaction for Scientific Machine Learning

## Abstract

Scientific machine-learning models often need physical or geometric constraints to hold over entire regions, not only at sampled points. We introduce mPOLICE, a general approach for imposing such constraints on multiple disjoint convex regions of ReLU networks. Its central geometric idea is to adjust the network so that each target region lies within its own affine activation cell. Once this local affinity is established, constraints on values, interfaces, output inequalities, and first derivatives that are linear in the local affine map reduce to finite vertex conditions and extend across the full regions. Distinct cells prevent distant constrained regions from being coupled through a shared affine map. We demonstrate the approach in operator learning, PDE boundary-value problems, geometric field reconstruction, and deterministic-policy collision avoidance. Across these applications, the returned models pass numerical verification of the imposed regional constraints and remain accurate on their primary prediction or control tasks. Periodic re-projection corrects constraint drift during continued training, while deployment uses the unchanged network architecture and incurs no additional inference-time computation.

## 1 Introduction

Scientific machine learning relies on neural networks to approximate PDE solutions, inverse maps, and operators, through physics-informed neural networks (PINNs) (Raissi et al., 2019; Karniadakis et al., 2021), variational methods such as Deep Ritz and variational PINNs (E & Yu, 2018; Kharazmi et al., 2019), and neural operators such as DeepONet and the FNO (Lu et al., 2021a; Li et al., 2021; Kovachki et al., 2023). In these settings, approximation accuracy alone is insufficient. Predictions must satisfy boundary conditions, interface conditions, conservation laws, and geometric constraints to remain physically meaningful, especially in applications such as multi-material diffusion, fluid dynamics, climate modeling, and inverse design, where even small violations can be qualitatively wrong (Lu et al., 2021b; Xie et al., 2024; Beucler et al., 2021).

Yet exact enforcement of such structure remains difficult. Most scientific-ML pipelines still impose boundary or interface conditions through soft penalties, whose weights are problem-dependent, can create challenging optimization landscapes, and provide no exact-arithmetic guarantee (Krishnapriyan et al., 2021). Existing hard-constraint methods provide important progress (Lu et al., 2021b; Xie et al., 2024; Beucler et al., 2021) but are often specialized to particular PDEs, coordinate systems, or boundary-condition forms. Neural operators need spatial constraints that transfer across unseen input functions, not merely approximate satisfaction on the training set (Lu et al., 2021a; Li et al., 2021; Kovachki et al., 2023). More broadly, standard deep learning methods often rely on sampling, penalties, or post-hoc corrections, with guarantees available only for particular constructions (Kotary et al., 2021; Kotary & Fioretto, 2024; Tordesillas et al., 2023).

Our approach exploits a geometric property of ReLU networks rather than relying on penalties. Such a network partitions its input space into polyhedral activation cells and restricts to one affine map on each cell. Constraint handling proceeds in two stages. *Affinity enforcement* places each constrained polytope inside one activation cell. *Constraint enforcement* imposes conditions linear in the resulting local affine map. Vertex values determine an affine function on a convex polytope, so constraints with affine residuals extend from the vertices to the full region. First derivatives on planar faces are represented through linear vertex

functionals after each face is thickened into a full-dimensional strip. The network architecture and inference graph remain unchanged. Throughout, *enforcement* denotes an algorithmic projection step. When the sign and vertex systems are feasible, the theorems provide provable exact-arithmetic guarantees.

POLICE (Balestriero & LeCun, 2023) solves the affinity-enforcement stage for a single convex region and has been applied where hard guarantees matter in robotics and control (Bouvier et al., 2024b;a). Realistic PDE settings, however, involve multiple disjoint boundary segments, interfaces, inclusions, and obstacles. A single activation pattern across all of them forces the network to be affine over their combined convex hull, coupling distant patches (Figure 1). Prior POLICE work also provided no general framework for the second stage of enforcing mixed equality, inequality, and derivative constraints on the resulting affine maps.

This paper addresses both stages. For affinity enforcement, mPOLICE extends POLICE with distinct region-wise activation patterns, so disjoint patches are not forced to share one affine map over their convex hull. For constraint enforcement, we give a common stacked-vertex formulation for value, inequality, coupled-interface, and first-derivative functionals, with derivative conditions handled on full-dimensional planar strips. We then integrate both projections into continued-training schedules with numerical verification that rejects any failed check. Although we motivate the framework through physical boundary and interface conditions, we show that the same mechanism applies to collision-avoidance constraints on deterministic policies, implicit-shape occupancy constraints, and operator-learning constraints.

Our contributions are threefold. First, building on the local-affinity principle used in prior work (Balestriero & LeCun, 2023; Tao et al., 2023; Tao & Thakur, 2024), we develop a continued-training workflow for independent multi-region constraints with projection and fail-closed verification. Second, we give a common stacked-vertex formulation for value, inequality, coupled-interface, and first-derivative functionals and instantiate it without changing the inference graph. Third, we evaluate the framework on operator learning, boundary-condition enforcement, implicit shape approximation, and deterministic policy learning with a numerically verified actor and a conditional exact-arithmetic collision-avoidance guarantee.

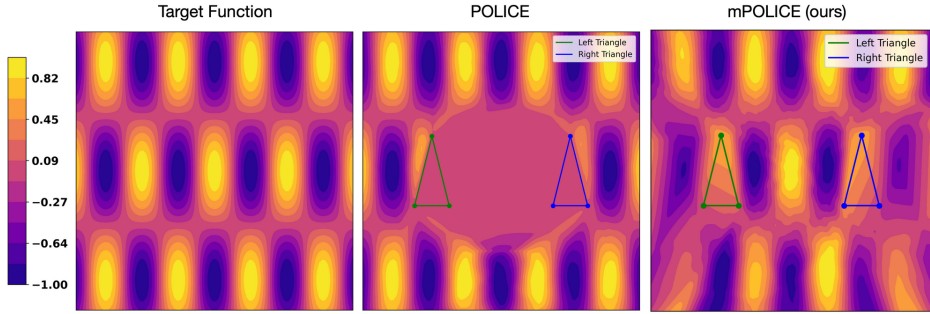

Figure 1: POLICE vs. mPOLICE. The left panel shows the target nonlinear field. The middle panel shows how one shared pattern on the two triangular regions forces affinity over their convex hull. The right panel shows how distinct realized patterns localize the forced affine behavior to separate containing cells.

## 1.1 Related Work

**Scientific machine learning.** PINNs (Raissi et al., 2019; Karniadakis et al., 2021), variational methods (E & Yu, 2018; Kharazmi et al., 2019), and neural operators (Lu et al., 2021a; Li et al., 2021; Kovachki et al., 2023) most commonly impose boundary and interface conditions through soft penalties, with tuning sensitivity and optimization difficulties (Krishnapriyan et al., 2021). Hard-constraint alternatives (Lu et al., 2021b; Xie et al., 2024; Beucler et al., 2021; Djeumou et al., 2022) tend to be specialized to particular PDEs, coordinate systems, or boundary-condition types, and neural operators additionally require spatial constraints that transfer to unseen input functions.

**Constrained learning beyond PDEs.** Architectural enforcement of monotonicity, convexity, output constraints, and differential conditions has been explored in several settings (Tordesillas et al., 2023; Kon-

stantinov et al., 2024; Zhong et al., 2023), and constraint-aware learning also arises in robotics, planning, and combinatorial optimization (Kondo et al., 2024; Bouvier et al., 2024b;a; Kotary et al., 2021; Kotary & Fioretto, 2024; Giannone et al., 2023; Picard et al., 2024). HardNet-Aff (Min & Azizan, 2024) appends a closed-form projection to every forward pass, so its hard affine output constraints hold wherever the network is evaluated. It does not establish that one affine map governs an entire input polytope, and its correction is paid at inference, whereas mPOLICE changes parameters before deployment. Neural Fields with Hard Constraints (Zhong et al., 2023) can impose differential conditions of arbitrary order through a specialized representation. Our narrower construction preserves an ordinary piecewise-linear network and therefore covers only constraints linear in its local affine pieces, principally values and first derivatives.

**Architecture-preserving V-polytope repair (APRNN).** APRNN (Tao et al., 2023; Tao & Thakur, 2024) is the most directly related prior work. Both methods modify network parameters without changing the architecture, control activation patterns so that each disjoint V-polytope lies in a locally affine piece, and thereby reduce whole-polytope output constraints to finite vertex relations. APRNN realizes these stages through its *Shift/Assert* sequence of joint LP/QPs. mPOLICE integrates multi-region assignment, repair, and verification with repeated projection during continued scientific-ML training. Experiment E8 compares one re-projection cycle on a four-hidden-layer, width-32 MLP with 4–256 vertices. mPOLICE takes 0.0040–0.0162 s and the APRNN/CVXPY-style adapter takes 0.173–4.539 s, giving a measured $43\times$–$280\times$ speedup. The adapter follows the released APRNN evaluation objective but uses CLARABEL/OSQP rather than the released Gurobi backend. The timings therefore characterize this controlled adapter protocol. Section 4.1 gives the full comparison, including the different objectives, anchors, and observed failure regimes.

**Relation to POLICE.** POLICE (Balestriero & LeCun, 2023) introduced activation-pattern control to enforce local affinity on a single convex region. mPOLICE extends this to multiple disjoint regions with independent activation patterns and develops the downstream equality, inequality, and derivative projections on vertex outputs. When the sign and vertex systems are feasible, the exact-arithmetic theorem extends those conditions over each region.

## 2 Methodology

We consider feedforward piecewise-linear deep networks, focusing on ReLU networks. This section formalizes the two stages of mPOLICE. *Affinity enforcement* (Sections 2.3–2.6) assigns and maintains region-wise activation patterns. The positive-margin condition provably makes the network affine on each designated region in exact arithmetic. *Constraint enforcement* (Sections 2.8–2.9) then uses linear systems on vertex outputs to impose target conditions.

### 2.1 Piecewise Affine Structure of ReLU Networks

A feedforward ReLU network defines a continuous piecewise-affine function. We use $L-1$ hidden ReLU layers followed by an affine output layer $L$. For $\ell = 1, \ldots, L-1$,

$$\boldsymbol{z}^{(\ell)} = \boldsymbol{W}^{(\ell)}\boldsymbol{x}^{(\ell)} + \boldsymbol{b}^{(\ell)}, \quad \boldsymbol{x}^{(\ell+1)} = \sigma(\boldsymbol{z}^{(\ell)}), \tag{1}$$

where $\boldsymbol{x}^{(1)} = \boldsymbol{x}$, $\sigma(u) = \max(u, 0)$, and $f_{\boldsymbol{\theta}}(\boldsymbol{x}) = \boldsymbol{W}^{(L)}\boldsymbol{x}^{(L)} + \boldsymbol{b}^{(L)}$. Within a fixed upstream pattern, every hidden pre-activation is affine in the network input, so its nondegenerate zero set is a hyperplane. Globally, deeper-layer boundaries are piecewise hyperplanar. The combined hidden signs induce finitely many activation-pattern regions (Montúfar et al., 2014). We call the set associated with one collection of weak branch inequalities a *closed activation-pattern region*. Such regions may overlap where pre-activations vanish. On each, the network agrees with an affine map $\boldsymbol{\Lambda}_r \boldsymbol{x} + \boldsymbol{\gamma}_r$. The argument extends to componentwise two-branch affine activations, such as Leaky ReLU, whose branches agree at zero. A nonlinear output activation would invalidate the vertex-to-region extension below.

## 2.2 Problem Setup and Preliminaries

Consider a deep neural network $f_{\boldsymbol{\theta}} : \mathbb{R}^D \to \mathbb{R}^K$ with parameters $\boldsymbol{\theta}$, representing a learned scalar or vector field, and $N$ disjoint convex polytopal regions $\{R_i\}_{i=1}^N \subset \mathbb{R}^D$ (in scientific-machine-learning applications, strips adjacent to boundary segments, material-interface neighborhoods, inclusions, electrodes, or other localized patches where exact structure is required), each described by a finite set of vertices $\{\boldsymbol{v}_p^{(i)}\}_{p=1}^{P_i}$. We write $[q] = \{1, \ldots, q\}$. Let

$$\boldsymbol{Y}_i(\boldsymbol{\theta}) = \begin{bmatrix} f_{\boldsymbol{\theta}}(\boldsymbol{v}_1^{(i)})^\top \\ \vdots \\ f_{\boldsymbol{\theta}}(\boldsymbol{v}_{P_i}^{(i)})^\top \end{bmatrix}, \qquad \boldsymbol{y}(\boldsymbol{\theta}) = \mathrm{col}_{i=1}^N \mathrm{vec}(\boldsymbol{Y}_i(\boldsymbol{\theta})). \tag{2}$$

Here "vec" stacks matrix columns and "col" concatenates vectors. Thus $\boldsymbol{y}(\boldsymbol{\theta}) \in \mathbb{R}^M$ for $M = K \sum_i P_i$. The finite constraint layer is stated most generally as

$$\mathcal{A}\boldsymbol{y}(\boldsymbol{\theta}) = \boldsymbol{a}, \qquad \mathcal{C}\boldsymbol{y}(\boldsymbol{\theta}) \le \boldsymbol{c}. \tag{3}$$

If the two systems contain $J_=$ and $J_\le$ rows, then $\mathcal{A} \in \mathbb{R}^{J_= \times M}$, $\boldsymbol{a} \in \mathbb{R}^{J_=}$, $\mathcal{C} \in \mathbb{R}^{J_\le \times M}$, and $\boldsymbol{c} \in \mathbb{R}^{J_\le}$. Rows of $\mathcal{A}$ and $\mathcal{C}$ may select one vertex, couple two vertices of a strip, or couple blocks from different regions. Thus Equation 3 includes spatially varying Dirichlet data sampled at vertices, paired interface conditions, and the cross-vertex difference quotients used for Neumann and Robin data. Extension between vertices requires the affinity hypothesis and an affine residual (Theorem 2), or a separately quantified approximation error for non-affine target data. A common pointwise special case is

$$\boldsymbol{E}_i f_{\boldsymbol{\theta}}(\boldsymbol{x}) = \boldsymbol{f}_i, \quad \forall \boldsymbol{x} \in R_i, \tag{4}$$

$$\boldsymbol{C}_i f_{\boldsymbol{\theta}}(\boldsymbol{x}) \le \boldsymbol{d}_i, \quad \forall \boldsymbol{x} \in R_i, \tag{5}$$

where $\boldsymbol{E}_i \in \mathbb{R}^{m_i \times K}$ and $\boldsymbol{f}_i \in \mathbb{R}^{m_i}$ specify $m_i$ constant equality constraints on the network output over $R_i$, and $\boldsymbol{C}_i \in \mathbb{R}^{q_i \times K}$ and $\boldsymbol{d}_i \in \mathbb{R}^{q_i}$ specify $q_i$ constant linear inequalities. These constraints raise two distinct questions. The first is how to guarantee that $f_{\boldsymbol{\theta}}$ is affine on *all* of $R_i$ rather than merely at sampled points. The second is how to enforce the desired constraints on the resulting affine map once affinity holds.

To answer the first question, we place each $R_i$ inside the closed activation-pattern region associated with one collection of weak branch inequalities, making $f_{\boldsymbol{\theta}}$ affine on $R_i$. Positive margins are additionally required for unambiguous cell interiors and the distinctness conclusion of Theorem 1 below:

$$f_{\boldsymbol{\theta}}(\boldsymbol{x}) = \boldsymbol{\Lambda}_i \boldsymbol{x} + \boldsymbol{\gamma}_i, \quad \boldsymbol{x} \in R_i, \tag{6}$$

where $\boldsymbol{\Lambda}_i \in \mathbb{R}^{K \times D}$ and $\boldsymbol{\gamma}_i \in \mathbb{R}^K$ are the slope and offset realized on the closed pattern region containing $R_i$, determined by $\boldsymbol{\theta}$ and its activation pattern. Consistent weak vertex signs give region-wise affinity. Strictly positive distinguishing margins establish pairwise cell distinctness.

We emphasize that local affinity is a *design mechanism*, not a modeling assumption. Equations 4–5 are semi-infinite constraints, and making $f_{\boldsymbol{\theta}}$ affine on $R_i$ is a sufficient condition reducing them to vertices. This restricts expressivity and leaves a best-affine approximation error for non-affine targets. Subdivision into separated convex pieces can lower interpolation error only when every sign subproblem remains feasible with positive realized margins. Regions assigned distinct patterns with positive margins cannot share boundary points; the gaps remain uncontrolled, and Experiment E7 in Appendix A shows no monotonic convergence guarantee.

Once affinity holds, the special case in Equations 4–5 need only be checked on the *finite* vertex set, replacing a continuum of conditions with finitely many linear relations:

$$\boldsymbol{E}_i (\boldsymbol{\Lambda}_i \boldsymbol{v}_p^{(i)} + \boldsymbol{\gamma}_i) = \boldsymbol{f}_i, \quad \text{or} \quad \boldsymbol{C}_i (\boldsymbol{\Lambda}_i \boldsymbol{v}_p^{(i)} + \boldsymbol{\gamma}_i) \le \boldsymbol{d}_i, \quad \forall p = 1, \ldots, P_i. \tag{7}$$

However, standard training offers no guarantee that $f_{\boldsymbol{\theta}}$ is affine on any specific $R_i$, nor that $R_i$ aligns with a single activation cell of the network's piecewise-affine decomposition. This is the primary obstacle to exact behavior on many physically distinct patches simultaneously.

### 2.3 From Single to Multiple Regions and the Convex Hull Problem

The original POLICE method (Balestriero & LeCun, 2023) solves the *affinity-enforcement* problem for a single convex region $R$. Given its vertices $\{\boldsymbol{v}_1, \ldots, \boldsymbol{v}_P\}$, it selects a binary sign pattern $\boldsymbol{s}^{(\ell)} = (s_1^{(\ell)}, \ldots, s_{N_\ell}^{(\ell)})$ for the $N_\ell$ neurons of each layer $\ell$ and adjusts the parameters to satisfy

$$0 \leq \min_{p \in [P]}(H_{p,k}^{(\ell)} s_k^{(\ell)}), \quad \forall k \in \{1, \ldots, N_\ell\}, \tag{8}$$

where $\boldsymbol{V}^{(\ell)}$ stacks the propagated vertex images as rows and $\boldsymbol{H}^{(\ell)} \triangleq \boldsymbol{V}^{(\ell)}(\boldsymbol{W}^{(\ell)})^T + \mathbf{1}_P(\boldsymbol{b}^{(\ell)})^T \in \mathbb{R}^{P \times N_\ell}$ is their pre-activation matrix. Here, $s_k^{(\ell)} \in \{-1, +1\}$ prescribes the branch of neuron $k$. Shared weak vertex signs place $R$ in the associated closed activation-pattern region and make $f_{\boldsymbol{\theta}}$ affine on $R$. When applied independently to multiple disjoint regions, however, POLICE may assign several of them the *same* pattern, making the network affine not just on their union but on their entire convex hull, the *convex hull problem*.

Imposing one shared pattern on several regions forces affinity over their convex hull. Alternatively, sequential independent bias-only repairs need not preserve the signs established for earlier regions. Appendix E gives one concrete infeasible pair of bias-only requirements. mPOLICE addresses this by assigning a unique global sign pattern to each region. When the assignment is feasible and its realized margins are strictly positive, Theorem 1 below proves that each $R_i$ lies in a distinct activation cell, avoiding the convex-hull coupling.

### 2.4 Multi-Region Sign Assignment and Parameter Adjustment

We formalize the assignment of unique sign patterns to the $N$ disjoint regions of a depth-$L$ ReLU network and the accompanying parameter adjustments, so that each $R_i$ is contained in a distinct activation cell. We introduce sign variables $\text{sign}_n^{(i,\ell)} \in \{+1, -1\}$, where $\ell \in \{1, \ldots, L-1\}$ indexes a hidden layer and $n$ a neuron in that layer. Their concatenation is the global pattern $\boldsymbol{S}_i$. The sign $\text{sign}_n^{(i,\ell)}$ prescribes the half-space

$$\text{sign}_n^{(i,\ell)} \left( \mathbf{w}_n^{(\ell)\top} \boldsymbol{v}_p^{(i,\ell)} + b_n^{(\ell)} \right) \geq \delta, \quad \forall p \in \{1, \ldots, P_i\}, \tag{9}$$

where $\boldsymbol{v}_p^{(i,\ell)}$ is the image of input vertex $\boldsymbol{v}_p^{(i)}$ after $\ell - 1$ hidden layers (it need not remain an extreme point), and $\delta \geq 0$ is the requested margin. The attained global margin is $m = \min_{i,\ell,n,\,p \in [P_i]} \text{sign}_n^{(i,\ell)}(\mathbf{w}_n^{(\ell)\top} \boldsymbol{v}_p^{(i,\ell)} + b_n^{(\ell)})$. When $\delta > 0$ is attained, each region lies strictly inside its prescribed activation-pattern region. With weak inequalities, affinity still holds, but patterns can be ambiguous at zero. Distinct strictly realized cells require distinct global patterns, and a positive margin on a distinguishing neuron is sufficient by Theorem 1.

Collecting these requirements yields a combinatorial, non-convex joint problem of minimizing a task objective $\Phi(\boldsymbol{\theta})$ subject to the *region-consistency* constraints above for all $p, i, n, \ell$ and the *uniqueness* constraints that no two regions share their global sign pattern. Appendix P gives a schematic program. In practice, we first choose $\text{sign}_n^{(i,\ell)}$ heuristically (Section 2.5) and then solve convex subproblems to enforce the assigned half-space constraints by adjusting $\{\mathbf{w}_n^{(\ell)}, b_n^{(\ell)}\}$ one layer at a time.

**Theorem 1** (Localized affine behavior). *Let $f_{\boldsymbol{\theta}}$ be a feedforward ReLU network with an affine output layer, and let $R_i = \text{conv}\{\boldsymbol{v}_1^{(i)}, \ldots, \boldsymbol{v}_{P_i}^{(i)}\}$.*

1. *If, for every hidden neuron and layer, the propagated vertices of $R_i$ have one common weak preactivation sign, then $R_i$ lies in the associated closed activation-pattern region and $f_{\boldsymbol{\theta}}$ is affine on all of $R_i$. The symbolic pattern need not be unique at zero pre-activations.*

2. *If this condition holds for every $R_i$ and, for each pair $i \neq j$, their prescribed global patterns differ at a neuron whose signed pre-activation has a strictly positive margin on both regions, then no single activation-pattern region contains both regions. Consequently, no shared-pattern condition forces one affine map on their convex hull.*

*In particular, pairwise-distinct prescribed patterns together with a global signed margin $m > 0$ satisfy the second condition.*

**Theorem 2** (Vertex-to-continuum residual bound). *Suppose $f_{\boldsymbol{\theta}}$ is affine on $R_i$ and $\boldsymbol{g}_i : R_i \to \mathbb{R}^m$ is affine. For any $\boldsymbol{E} \in \mathbb{R}^{m \times K}$ and any norm on $\mathbb{R}^m$,*

$$\sup_{\boldsymbol{x} \in R_i} \|\boldsymbol{E} f_{\boldsymbol{\theta}}(\boldsymbol{x}) - \boldsymbol{g}_i(\boldsymbol{x})\| \leq \max_{p \in [P_i]} \left\| \boldsymbol{E} f_{\boldsymbol{\theta}}(\boldsymbol{v}_p^{(i)}) - \boldsymbol{g}_i(\boldsymbol{v}_p^{(i)}) \right\|. \tag{10}$$

*Likewise, for $\rho \geq 0$, if an affine inequality residual is at most $\rho$ componentwise at every vertex, it is at most $\rho$ throughout $R_i$. The same argument applies on a boundary face to any residual affine in its face coordinates, including the strip-derived derivative functionals in Appendix D.*

Both theorems are exact-arithmetic results proved in Appendix C.

## 2.5 Strategies for Sign Assignment

We use three heuristic rules. *Majority voting* sets $\mathrm{sign}_n^{(i,\ell)} = +1$ if the majority of the vertex pre-activations $z_n^{(\ell)}(\boldsymbol{v}_p^{(i,\ell)}) = \mathbf{w}_n^{(\ell)\top} \boldsymbol{v}_p^{(i,\ell)} + b_n^{(\ell)}$ are positive and $-1$ otherwise. Exact ties are assigned positive. The *mean rule* instead thresholds the vertex average $\bar{z}_n^{(i,\ell)} = \frac{1}{P_i} \sum_{p=1}^{P_i} z_n^{(\ell)}(\boldsymbol{v}_p^{(i,\ell)})$ at $\tau$. Every experiment uses $\tau = 0$, leaving ambiguous neurons to enforcement at margin $\delta$ and subsequent numerical verification. We reserve $\delta$ for the enforcement margin and $\varepsilon$ for the stopping tolerance of Figure 2. The superscript in the thickened strip $R^\varepsilon$ of Section 3 is a geometric thickness.

*Max-margin error-correcting output-code (ECOC) assignment* builds uniqueness in from the start. Provided at least $b = \lceil \log_2 N \rceil + r$ candidate neurons are available, the $b$ neurons with the largest worst-region current best-direction signed margins become *identifier bits* (redundancy $r = 1$ in our experiments), and a greedy matching assigns distinct codewords close in Hamming distance to the natural signs. Remaining neurons keep their larger current signed margin per region. This guarantees symbolic uniqueness under that capacity condition but only *aims* to reduce enforcement movement. It is not a global movement minimizer.

The mean rule works well when pre-activations cluster. Majority voting resists small outlier sets but can flip when a region straddles a boundary. ECOC buys uniqueness by construction at the cost of a few prescribed bits. Under the first two rules, duplicate global patterns (typically from nearby regions) are broken by flipping a few near-zero-pre-activation neurons in one region. Since a flip can itself create a duplicate, uniqueness is re-checked rather than assumed (the pairwise verification in Algorithm 1).

**Feasibility, detection, and repair.** A prescribed sign assignment need not be feasible. For a first-layer neuron the pre-activation is affine in the input, so the pattern $(+, -, +)$ across three collinear regions is unrealizable by any $(\mathbf{w}, b)$. Because enforcement operates neuron-by-neuron (Section 2.6), a returned assignment failure is localized by the signed margin $m$. Theorem 1 is conditional on $m > 0$, so any returned point that fails to realize the assigned signs is rejected. However, $m \leq 0$ does not by itself prove that the half-space system is infeasible. Repair uses a heuristic sign re-assignment. Each violating neuron flips its least-committed region's sign (smallest $|\bar{z}_n^{(i,\ell)}|$), uniqueness is re-verified, and enforcement is re-run. A colliding pattern is resolved by a distinguishing flip. Repeated flips, restarts, or added capacity may be required. There is no completeness guarantee, and the failure is reported explicitly if no attempt passes verification. The data-driven rules above select signs the pretrained network already realizes, which favors feasibility but does not guarantee it. Algorithm 1 applies this bounded verification-and-repair procedure after enforcement. Experiment E12 in Appendix B demonstrates a complete detect-and-repair cycle on the collinear construction.

## 2.6 Enforcing Multi-Region Affinity via Sign Patterns

Once the region-wise signs and propagated vertices for a layer are fixed, maintaining them is a standard projection onto margin-separated half-spaces. mPOLICE constructs, repairs, and verifies the distinct multi-region assignment around this projection. These operations establish the localized affine premise but do not themselves impose downstream constraints.

We implement three standard backends, with their formulations, geometry, and selection discussed in Appendix O. A *per-neuron QP* gives the minimal parameter change per neuron to solver tolerance. *POCS*

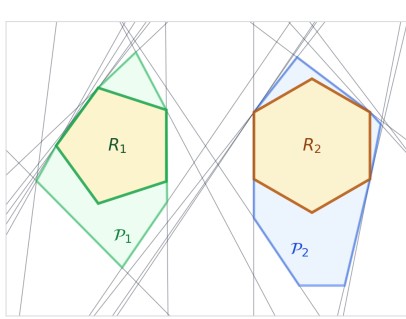
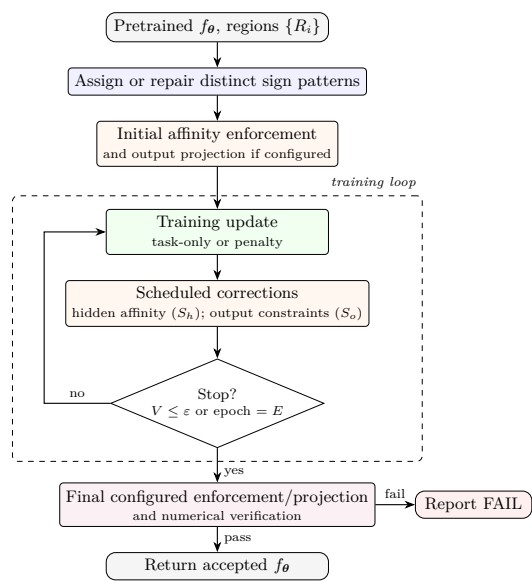

Figure 2: Left: target regions (yellow) within activation cells (shaded). Right: after sign assignment and enforcement, mPOLICE alternates training and projections, then corrects and verifies the model or reports failure (Algorithm 1).

repeatedly projects onto violated half-spaces and is factorization-free. *Batched ADMM* (Gabay & Mercier, 1976) alternates a quadratic update with projection onto the shifted orthant, shares one factorization across a layer's neurons and iterations, and supports $\delta = 0$. All three address the *same* convex feasible set, so any returned point that passes the common margin check supports the same affinity conclusion. Finite iteration budgets need not reach such a point. Only the QP has a per-neuron minimal-change guarantee, while POCS and iteration-capped ADMM can return different feasible points. Backends can also warm-start one another. The POCS→QP hybrid runs inexpensive sweeps before one QP projection from the POCS iterate. Appendix P details time complexity and termination.

## 2.7 Target Region vs. Activation Cell

When the positive-margin condition is verified, Theorem 1 guarantees that each target region $R_i$ lies in a single activation cell, but that cell can be strictly larger because its facets are network ReLU boundaries rather than facets of $R_i$ (Figure 2). Neither increasing width nor continuing training guarantees that this cell shrinks. Enforcement can move its boundaries either way. Experiment E10 in Appendix B reports an empirical width trend for one configuration, not a monotonic theorem. Exact matching would require the relevant facets to coincide with active ReLU boundaries, but the mismatch does not weaken the provable affine result on $R_i$ itself.

## 2.8 Training with Separate Affinity Enforcement and Constraint Satisfaction

After sign-pattern assignment (Section 2.5) and enforcement (Section 2.6), the network is affine on each $R_i$. Specifically, $f_{\boldsymbol{\theta}}(\boldsymbol{x}) = \boldsymbol{\Lambda}_i \boldsymbol{x} + \boldsymbol{\gamma}_i$ for $\boldsymbol{x} \in R_i$. An affine map on a convex polytope is determined on the polytope by its vertex values. For a *full-dimensional* region (dim $R_i = D$), choose an index set $\mathcal{I}_i \subset [P_i]$ of $D + 1$ affinely independent vertices. Let $B_i \in \mathbb{R}^{(D+1) \times (D+1)}$ stack their augmented coordinates and let $\boldsymbol{Y}_{i,\mathcal{I}_i} \in \mathbb{R}^{(D+1) \times K}$ contain the corresponding rows of $\boldsymbol{Y}_i$. Then $(\boldsymbol{\Lambda}_i \; \boldsymbol{\gamma}_i)^\top = B_i^{-1} \boldsymbol{Y}_{i,\mathcal{I}_i}$.

For a lower-dimensional region with dim $R_i = k_i < D$, no ambient inverse exists. Vertex values pin the map only on the affine hull, so they determine output and tangential constraints but not ambient normal derivatives. We therefore impose derivative conditions on full-dimensional thickened strips (Section 3). Experiment E13 in

Appendix B studies lower-dimensional simplex patches, while the strips and pads elsewhere are full-dimensional but thin.

The construction need not form $B_i^{-1}$. Local and cross-region conditions are assembled directly in the stacked form of Equation 3. The construction is exact only when the desired functional is determined by the available vertex outputs. In particular, an ambient normal derivative is not determined by a zero-thickness, lower-dimensional face alone.

Throughout the paper, *pretraining* denotes task-only training of the unconstrained network. *Fine-tuning* denotes the constrained continuation after sign assignment. Depending on the experiment, this continuation uses the penalty loop below, output-layer-only training with periodic re-projection (Section 2.9), or training restricted to a parameter subset. DeepONet, for example, fine-tunes the branch and trunk output layer. The DeepONet and 3D-shape studies also use task-specific geometry-guided preconditioning before affinity enforcement.

The training loop (Figure 2, right, and Algorithm 1) starts from a pretrained $f_{\boldsymbol{\theta}}$, enforces the assigned sign patterns (Section 2.6), and then alternates epochs of $\mathcal{L}_{\text{task}} + \lambda \mathcal{L}_{\text{con}}$ minimization ($\mathcal{L}_{\text{con}}$ accumulates vertex-constraint penalties across regions) with sign re-enforcement. We measure stacked-constraint violation:

$$V(\boldsymbol{\theta}) = \max\{\|\mathcal{A}\boldsymbol{y}(\boldsymbol{\theta}) - \boldsymbol{a}\|_\infty,\ \|[\mathcal{C}\boldsymbol{y}(\boldsymbol{\theta}) - \boldsymbol{c}]_+\|_\infty\},$$

where $[\cdot]_+$ is applied componentwise and an absent constraint block is omitted. In the optional early-stopping mode, training stops when $V(\boldsymbol{\theta}) \leq \varepsilon$, where $\varepsilon$ is a user-specified task-scale heuristic rather than a certification tolerance (cf. Appendix P). The reported experiments do not use this condition to terminate. They use fixed epoch or step budgets and separate final verification gates. The value of $\lambda$ increases when violations persist past a patience window, and the best model by total loss is restored at completion.

## 2.9 Projected Training and Output-Layer Projection

As an alternative to the penalty-based loop, we implement a projected training variant. Each epoch performs $T_{\text{SGD}}$ steps of task-only SGD ($\boldsymbol{\theta} \leftarrow \boldsymbol{\theta} - \alpha\nabla\mathcal{L}_{\text{task}}$, with learning rate $\alpha$ and $\eta$ reserved for the constraint-tightening margin), then monitors the minimum signed margin across all regions. When this margin drops below a threshold $\delta_{\text{trigger}}$, parameters are re-projected onto the sign-feasible half-spaces via QP, POCS, or ADMM. With hidden features fixed, Equation 3 is linear in the vector $\boldsymbol{\theta}_o$ of final-layer weights and biases. For a *consistent* equality system $A\boldsymbol{\theta}_o = c$, the Euclidean minimal-change projection is

$$\boldsymbol{\theta}_o^\star = \boldsymbol{\theta}_{o,0} - A^\top (AA^\top)^\dagger (A\boldsymbol{\theta}_{o,0} - c). \tag{11}$$

Here $^\dagger$ is the Moore–Penrose pseudoinverse. Consistency is checked through the post-solve residual (and, where reported, rank diagnostics). If $A\boldsymbol{\theta}_o = c$ is inconsistent, Equation 11 returns only a least-squares residual correction, not a feasible projection, and the post-solve residual check rejects it. Inequalities use a constrained QP or another backend followed by the same feasibility check. For equalities and $g = \nabla_{\boldsymbol{\theta}_o}\mathcal{L}_{\text{task}}$, $g_{\text{proj}} = g - A^\top (AA^\top)^\dagger (Ag)$ is the orthogonal projection onto the null space $\ker A = \{v : Av = 0\}$ and preserves the feasible affine space in exact arithmetic for fixed $A$. This tangent formula alone does *not* preserve inequalities. Those require a feasible step in the tangent cone (including a suitable step size) or explicit re-projection. Floating-point drift and changes to hidden features likewise require re-projection and renewed numerical verification.

Algorithm 1 summarizes the workflow. Starting from a pretrained network, mPOLICE assigns—and, if needed, repairs—a distinct sign pattern for each region, enforces the patterns in the hidden layers, and projects the output layer onto the vertex constraints when configured. During continued training, task-only or penalty updates alternate with scheduled hidden-layer re-enforcement and output-layer re-projection. Training stops when $V(\boldsymbol{\theta}) \leq \varepsilon$ or the epoch budget is exhausted. After any configured final projections, the algorithm checks margins, uniqueness, residuals, and finiteness. It returns the model and measured values if those checks pass. Otherwise, it reports failure. Verification-off runs are labeled diagnostic.

Appendix B provides the full pseudocode, retry rules, schedule and verification definitions, and exact per-experiment settings in Tables 10 and 11.

Table 1: Spiral benchmark on two disjoint inequality-constrained disks (mean $\pm$ std over 3 seeds; bold: best mean; scaling in headers). Training uses 400 pretraining epochs plus 200 continuation epochs for constrained methods. The last column times one 4096-query batch.

| Method | MSE Out ($\times 10^{-4}$) | MSE All ($\times 10^{-4}$) | Max Viol. ($\times 10^{-2}$) | Affine RMSE ($\times 10^{-2}$) | Train (s) | 4096-query batch (ms) |
|---|---|---|---|---|---|---|
| Unconstrained | $11.95 \pm 1.94$ | $11.53 \pm 2.05$ | $8.89 \pm 1.29$ | $3.91 \pm 0.11$ | $\mathbf{12.97 \pm 0.74}$ | $\mathbf{0.10 \pm 0.00}$ |
| Soft Penalty ($\lambda{=}1$) | $7.92 \pm 1.69$ | $7.74 \pm 1.56$ | $1.19 \pm 0.38$ | $3.22 \pm 0.05$ | $25.00 \pm 1.30$ | $\mathbf{0.10 \pm 0.00}$ |
| Soft Penalty ($\lambda{=}10$) | $8.31 \pm 1.74$ | $8.19 \pm 1.63$ | $0.28 \pm 0.34$ | $3.13 \pm 0.10$ | $24.98 \pm 1.29$ | $\mathbf{0.10 \pm 0.00}$ |
| Soft Penalty ($\lambda{=}100$) | $10.06 \pm 2.65$ | $10.02 \pm 2.48$ | $0.03 \pm 0.04$ | $3.05 \pm 0.17$ | $24.97 \pm 1.31$ | $\mathbf{0.10 \pm 0.00}$ |
| HardNet-Aff | $\mathbf{7.81 \pm 1.77}$ | $\mathbf{7.44 \pm 1.66}$ | $\mathbf{0.00 \pm 0.00}$ | $3.69 \pm 0.03$ | $20.65 \pm 1.02$ | $0.20 \pm 0.00$ |
| mPOLICE-POCS | $75.48 \pm 35.13$ | $73.00 \pm 33.58$ | $\mathbf{0.00 \pm 0.00}$ | $\mathbf{0.00001 \pm 0.00000}$ | $19.68 \pm 0.84$ | $\mathbf{0.10 \pm 0.00}$ |

## 3 Constraint Families Across Domains

Once affinity holds on $R_i$, vertex-determined constraints linear in $(\boldsymbol{\Lambda}_i, \boldsymbol{\gamma}_i)$ reduce to Equation 3. This covers values, coupled interface conditions, output inequalities, and first derivatives on full-dimensional planar strips $R^\varepsilon$. Appendix D gives the Neumann and Robin reductions, their continuum assumptions, and the $1/\varepsilon$ conditioning bound.

## 4 Experiments

The supplementary package contains the mPOLICE library and result-reproduction scripts. Except for the task-specific preconditioning described in Section 2.8, the mPOLICE configurations instantiate Algorithm 1. Tables 10 and 11 record their assignment rules, solvers, margins, schedules, and verification modes. Unless stated otherwise, experiments use float64. The spiral benchmark and returned RL actor use float32.

**Metrics.** Unless a table states otherwise, equality violation is the maximum absolute scalar residual over constraint rows, regions, and the specified evaluation points, and inequality violation is $\max_{i,s,j}[\boldsymbol{C}_i f_{\boldsymbol{\theta}}(\boldsymbol{x}_s) - \boldsymbol{d}_i]_j^+$. Comparative dense-sample metrics use the same evaluation points for every method. Vertex residuals are reported separately as numerical-verification quantities. The DeepONet table averages per-forcing metrics over 50 test forcings from one trained model, whereas Robin reports the mean squared boundary-functional residual. For dense samples $\{\boldsymbol{x}_s\}_{s=1}^S \subset R_i$, we define

$$\text{Affine RMSE}(R_i) \;=\; \min_{\boldsymbol{A} \in \mathbb{R}^{K \times D},\, \boldsymbol{c} \in \mathbb{R}^K} \sqrt{\tfrac{1}{S} \sum_{s=1}^S \left\| f_{\boldsymbol{\theta}}(\boldsymbol{x}_s) - (\boldsymbol{A}\boldsymbol{x}_s + \boldsymbol{c}) \right\|_2^2}, \tag{12}$$

the residual of the least-squares best affine fit to the network on $R_i$. It equals zero exactly when one affine map fits every sampled output. This metric separates *region-wise affinity*, which mPOLICE enforces, from *pointwise feasibility*, which methods like HardNet-Aff provide.

### 4.1 Baseline Comparisons

Before turning to physics applications, we isolate what distinguishes mPOLICE from existing constraint-enforcement methods. We benchmark against soft penalties and HardNet-Aff (Min & Azizan, 2024) on a 2D spiral regression task with two disjoint sign-constrained disks, placed where the target field violates the required signs so that enforcement genuinely conflicts with the data. Every arm starts from the same pretrained checkpoint. The unconstrained row is that pretrained reference, while the four constrained arms share the same additional 200-epoch fine-tuning budget. Appendix B gives the protocol, precision, and metric details. We use mPOLICE-POCS here and benchmark all three backends in Section 4.5.

Table 1 illustrates the feasibility-versus-affinity gap on this benchmark without positive-margin verification. Over the tested penalty weights, soft penalties reduce the dense maximum violation to $3 \times 10^{-4}$ but remain non-affine on the disks (Affine RMSE $\approx 3.1 \times 10^{-2}$). HardNet-Aff has zero measured violation and nearly preserves the unconstrained task error, but its Affine RMSE ($3.69 \times 10^{-2}$) remains close to the unconstrained model's ($3.91 \times 10^{-2}$). Its projection also runs at each forward call, doubling the 4096-query batch latency here ($0.20$ vs. $0.10\,\text{ms}$). mPOLICE-POCS instead drives sampled Affine RMSE to the float32 floor ($\approx 10^{-7}$)

and has zero measured dense violation with an unchanged inference graph, but at substantially higher task MSE. Thus region-wise affinity is an enabling restriction for derivative and local-map constraints, not an accuracy advantage for simple output inequalities that HardNet-Aff already handles well.

The higher MSE is the observed price of forcing one affine map on a containing cell that may exceed the target polygon. Experiment E10 in Appendix B quantifies this protocol-specific trade-off. Increasing $\delta$ enlarges the sampled cell/region ratio from 1.21 to 2.30 while outside MSE rises from $9.6 \times 10^{-3}$ to $1.6 \times 10^{-2}$. Increasing width from 16 to 128 instead reduces the observed ratio to 1.04 and the constrained/unconstrained MSE ratio from 7.8 to 2.2. These are empirical trends, not monotonic guarantees. Experiment E11 in the same appendix assigns one shared pattern to both disks. It produces one cell of area $\approx 3.4$ rather than mPOLICE's two cells of area $\approx 0.54$. The resulting model is affine across the inter-disk hull and has $19\times$ higher outside MSE.

**Comparison with APRNN.** APRNN (Tao et al., 2023) is the closest hard-constraint baseline and shares the V-polytope reduction. Appendix A reports Experiments E0–E8 using an APRNN/CVXPY-style adapter that follows the released evaluation objective and is checked stage by stage against the released implementation. Both workflows reach solver-scale Robin residuals in E1. Their anchors and training protocols differ, so the fit values describe the configured workflows rather than an isolated accuracy comparison. The E2 re-projection variants finish with poor relative error of 0.99–1.05. In E8, mPOLICE is $43\times$–$280\times$ faster per re-projection cycle. APRNN solves joint LP/QPs whose constraint count grows with the vertex count $V$. mPOLICE's ADMM instead pays $O(Vn^2 + n^3)$ once per layer and reuses the factorization across $W$ neurons and $J$ iterations, whose updates cost $O(JWn(V+n))$. The timing comparison uses the CLARABEL/OSQP adapter rather than the released Gurobi backend. The appendix also reports the failure regimes for both methods.

### 4.2 Neural Operator with Internal Dirichlet Constraints

We apply mPOLICE to a DeepONet (Lu et al., 2021a) learning the Poisson solution operator on $[0,1]^2$ with two disjoint internal Dirichlet pads $R_1, R_2$, where $u = 0$. Writing $u_\theta(f, y) = \mathbf{b}(f)^\top \mathbf{t}(y)$, mPOLICE constrains only the trunk $\mathbf{t}$. Enforcing trunk affinity on each pad and projecting its vertex outputs to zero yields, under exact arithmetic,

$$u_\theta(f, y) = \mathbf{b}(f)^\top \mathbf{0} = 0 \quad \forall f, \ \forall y \in R_1 \cup R_2. \tag{13}$$

Numerical verification finds distinct positive-margin patterns and a maximum trunk vertex residual of $1.943 \times 10^{-16}$. The exact-arithmetic guarantee is forcing-independent, whereas finite-precision bounds scale with the branch norm; on 50 test forcings, the conditional bound is $9.8 \times 10^{-17}$ and is not uniform over the forcing space. On these unseen forcings, mPOLICE attains 8.8% relative error and a mean maximum violation of $2.43 \times 10^{-17}$, about twelve orders below the best tested soft variant. Figure 3 and Appendix Table 19 summarize the results. This run freezes the hidden trunk; Experiment E2 in Appendix A examines continued trunk training with re-projection.

### 4.3 Robin Boundary Condition

We show mPOLICE on steady conduction over $[0,1]^2$ with Dirichlet data on three sides and a planar Robin segment $hT + k\partial_n T = hT_{\text{amb}}$ at $x = 0$, $y \in [0.3, 0.7]$. The wall is thickened into a strip. ADMM enforces affinity, a tighter QP refines it at margin $10^{-4}$, and output projection imposes the Robin vertex functional. Numerical verification shows a minimum signed margin of $1.3 \times 10^{-4}$ and maximum Robin functional residual of $2.9 \times 10^{-15}$. Figure A6 shows Robin MSE $1.3 \times 10^{-30}$ and interior MSE $1.4 \times 10^{-5}$ for this run. Across the tested $\lambda \in \{1, 10, 100, 1000\}$, the best soft Robin MSE is $2.4 \times 10^{-6}$, with increasing weights trading wall residual against interior fit.

### 4.4 Certified Deterministic Collision Avoidance

Figure 4 shows a TD3 navigation example with a numerically verified deterministic deployed policy and a conditional exact-arithmetic guarantee. Each black line obstacle is enclosed by the closed red clearance rectangle obtained by expanding it by 0.01 in both coordinates. After checkpoint serialization and reload,

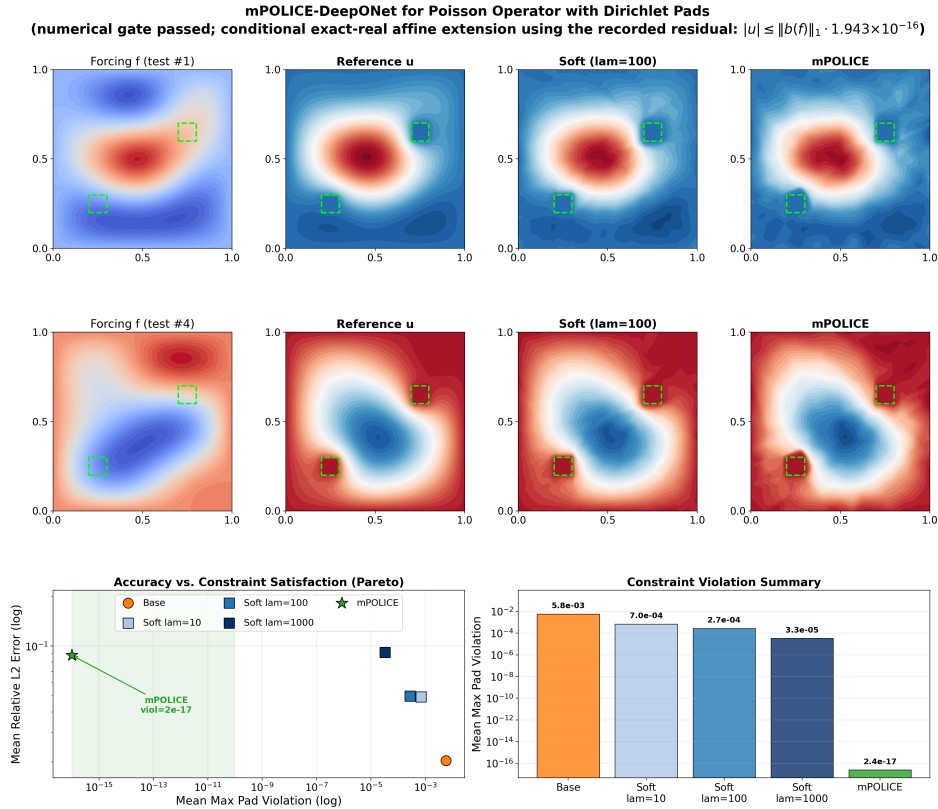

Figure 3: DeepONet with mPOLICE: predictions for two test forcings (top; green dashed squares mark the internal pads) and the accuracy–violation trade-off (bottom). The numerical bound assumes the reported floating-point residual upper-bounds its exact-arithmetic counterpart.

numerical verification of the returned float32 actor reports a minimum hidden signed margin of $1.4997 \times 10^{-3}$ and a minimum raw output-inequality slack of $1.9999 \times 10^{-3}$. This leaves $9.9986 \times 10^{-4}$ beyond the required $10^{-3}$ output margin. Under the exact-arithmetic conditions proved in Appendix G, every transition segment of the clipped deterministic actor remains outside the protected rectangles from any in-domain initial state outside them, and therefore cannot intersect either obstacle. The 100-start deterministic evaluation has a mean reward of 4.51 and zero observed collisions.

The appendix also contains a Neumann strip, geometry-guided 3D occupancy, synthetic field regression, non-convex decomposition, and a partially covered internal wall. The wall example studies coverage explicitly. Its strips cover $\approx 96\%$ of the boundary with measured covered flux below $1.5 \times 10^{-13}$, while the uncovered gaps have order-one measured flux. The unequal-fit soft comparison is qualitative.

## 4.5 Computational Performance

We time complete, numerically verified calls on a frozen four-hidden-layer, width-64 ReLU MLP for 1, 2, and 4 regular-polygon regions (32, 64, and 128 total vertices) over 3 seeds. All 27 runs pass. At 4 regions, mean totals—including assignment, affinity enforcement, output projection, and verification—are $0.025\,\mathrm{s}$ (POCS), $0.806\,\mathrm{s}$ (ADMM), and $0.894\,\mathrm{s}$ (QP). All three retain $\approx 3.3\,\mathrm{ms}$ latency per 4096-query batch, so backend choice affects enforcement rather than deployment. Figure A1, Table 22, and Appendix N give the phase breakdown and protocol. Vertex-count sweeps reveal different backend scaling. With two disks, increasing vertices per disk from $V = 16$ to $1024$ (32–2048 total) raises POCS time only from 0.008 to $0.014\,\mathrm{s}$, while capped ADMM solves range from 0.004 to $0.049\,\mathrm{s}$; per-neuron QP grows from 0.010 to $5.620\,\mathrm{s}$. For $P$ total vertices, layer input width $n$, $W$ neurons, and fixed $J$ iterations, POCS costs $O(JPWn)$ per layer. ADMM costs

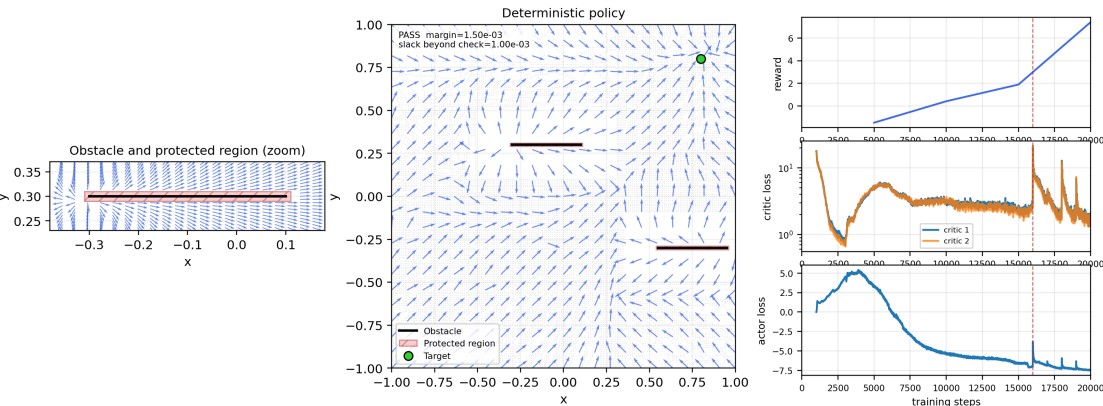

Figure 4: Verified deterministic TD3 policy with a conditional exact-arithmetic collision-avoidance guarantee. Left: upper obstacle (black), its 0.01 protected rectangle (red), and policy directions. Center: normalized directions for both obstacles and target. Right: TD3 training, with late affinity enforcement marked by the dashed line. See Appendix G.

$O(Pn^2 + n^3 + JWn(P + n))$ but reuses one factorization across neurons, consistent with its flatter measured scaling; QP instead makes $W$ solver calls and grows superlinearly in this sweep. E8 provides a separate per-cycle comparison on a width-32 network. As one region grows from 4 to 256 vertices, mPOLICE rises from 0.0040 to 0.0162 s, whereas the APRNN/CVXPY-style adapter rises from 0.173 to 4.539 s, a measured $43\times$–$280\times$ gap.

**Appendix roadmap.** Appendices A and B report controlled Experiments E0–E8 and E9–E13, respectively, with the latter also giving the phase-template algorithm and configuration tables. Appendix C proves the two theorems, Appendix D derives the planar-strip functionals, Appendix E gives the bias-only contradiction, and Appendix F discusses non-convex decomposition. Appendices G, H, I, J, K, L, and M provide the RL, DeepONet, Neumann, partially covered wall, implicit-shape, Robin, and synthetic-field details, respectively. Appendix N documents the performance benchmark, Appendix O gives the backend formulations and selection guidance, and Appendix P gives the complexity, termination, and verification analysis.

## 5 Limitations

The current implementation and analysis target fully connected ReLU or two-slope Leaky ReLU MLPs and conditions whose residuals are affine in the resulting local map (Appendix D); other architectures or nonlinear and second-order conditions require new machinery. Enforcing affinity can constrain an activation cell larger than the target region, so low MSE outside the constrained regions is not guaranteed and depends on the task, geometry, architecture, and feasible assignments. Experiments E10–E13 in Appendix B characterize these trade-offs and feasibility limits, while Appendix P details the verification scope.

## 6 Conclusion

mPOLICE provides an architecture-preserving framework for imposing independent value, interface, inequality, and first-derivative constraints over multiple convex regions of piecewise-linear MLPs. Distinct-cell affinity, stacked vertex functionals, and training-time re-projection reduce whole-region requirements to finite conditions while leaving the deployed inference graph unchanged. Across operator learning, PDE boundary conditions, geometric field reconstruction, and deterministic control, the experiments demonstrate numerically verified constraint satisfaction together with the theorems' conditional exact-arithmetic guarantees. These results establish a practical foundation for extending region-wise hard constraints to broader architectures and scientific models.

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

# A    APRNN Comparison Experiment Details

This appendix gives the detailed APRNN-vs.-mPOLICE evidence summarized in Section 4.1. Runs labeled *APRNN/CVXPY-style* implement APRNN's Shift/Assert formulation but solve the LP/QPs through CLARABEL/OSQP rather than the released Gurobi backend. E1 and E2 are smaller-scale variants of the main Robin and DeepONet problems. We use reduced architectures because the APRNN/CVXPY-style joint LP/QPs at the paper-scale architectures were too slow for these complete controlled workflows, especially E2's initial repair followed by 20 periodic re-projections. This is a workflow-level design choice, not a claim that a single paper-scale call is impossible: the architecture-scale diagnostics below time one simplified, randomly initialized, one-strip invocation. E0 verifies the adapter against released `sytorch`+Gurobi on small instances. E3–E8 study drift, qualitative inequalities, scaling, boundaries, and re-projection. E1–E3 use seed 3, and E4 uses seed 7. Later tables state when they average three seeds. The reported APRNN wall times characterize the adapter and may differ under the released APRNN/Gurobi implementation.

## E0. Adapter Verification

APRNN Algorithm 1 uses a sequence of Shift calls on network *slices* $N[k_i : \ell_i]$ followed by a final Assert call. Inputs are forwarded concretely through layers $[0, k_i)$ and symbolically through the slice. The first symbolic layer carries symbolic weights and biases. Subsequent symbolic layers carry concrete weights and symbolic biases, and all ReLU activation patterns are pinned to the centroid reference pattern. The adapter implements this slice structure. Its default partition uses consecutive single-layer slices and places Assert on the output layer. E0 compares it with released `sytorch`+Gurobi on small instances.

The APRNN/CVXPY-style adapter follows the objective implemented in the released `eval_3_aprnn.py` script at APRNN commit `5f676a5`. It minimizes the sum of separately normalized $\ell_\infty + \ell_1/n$ terms for the parameter and output deltas. During each Shift stage, the output delta is measured against the corresponding prefix output of the original network; the final Assert uses the original full-network output. APRNN Algorithm 1 instead constructs one concatenated output-and-parameter vector $\Delta$ and minimizes $\|\Delta\|_p$. Our experiments follow the executable `eval_3` objective for this choice; they do not otherwise reproduce the `eval_3` protocol. Verification uses a `sytorch`+Gurobi implementation of this objective configuration, driven through the same primitives as `eval_3`. For the single-hidden-layer network (1-D $\to 4 \to$ 1-D), both implementations select the same centroid patterns and satisfy the vertex specifications at solver precision. The two-hidden-layer network (1-D $\to 4 \to 4 \to$ 1-D) covers three cases. They are the multi-slice partition $s = ((0, 2), (2, 4))$ with two Dirichlet polytopes, a *non-determining* single-vertex inequality specification, and a Shift slice spanning two affine layers. In the non-determining case, the feasible set does not pin down the function, so agreement validates the optimization objective rather than the constraint set. In all cases the implementations agree stage-by-stage in every layer's returned parameters to $\leq 1.8 \times 10^{-6}$ and pointwise (inside and outside the polytopes) to $\leq 1.6 \times 10^{-6}$, with specifications satisfied at solver precision and exact local affinity. The adapter pins activation patterns with margin $10^{-6}$, while `sytorch` uses non-strict constraints at 0. The remaining $\sim 10^{-6}$ differences reflect this margin together with solver tolerances. E1, E3, E4, E5, E6, and E8 also use supervised task anchors. Anchors are propagated symbolically only through slices without ReLUs, where the linearization is exact. The experiments also use cross-vertex linear functionals in E1 and per-vertex target values within one region in E7. Published APRNN uses specifications that are uniform over each polytope. The *APRNN/CVXPY-style* label denotes this released-objective adapter together with those extensions. An architecture-scale adapter diagnostic takes 1.095 s for a width-128, two-hidden-layer trunk with 32 outputs and 10.237 s for a width-256, three-hidden-layer scalar-output network. These calls use randomly initialized networks with one four-vertex strip, rather than the trained checkpoints or complete DeepONet and Robin workflows.

## E1. Robin Boundary Condition (smaller-scale variant of Section 4.3)

A two-hidden-layer, width-32 coordinate ReLU MLP, smaller than the three-hidden-layer, width-256 MLP used in Section 4.3, fits the same finite-difference reference for $-\Delta T = Q$ on $[0, 1]^2$ with Dirichlet data on three sides and the same Robin condition $hT + k\partial_n T = hT_{\text{amb}}$ on the same cooling-wall segment. The

Robin law reduces, exactly as in the main paper, to two linear functionals on the strip-vertex outputs. Both APRNN/CVXPY-style and mPOLICE receive the same reduction.

Table 2: E1, Robin condition. Both hard methods reach numerical precision with comparable interior fit. APRNN uses one repair with 512 anchors at weight 1, while mPOLICE uses projected training with all anchors at weight 100 plus output-layer fine-tuning. The interior-MSE column therefore compares complete workflows, not isolated enforcement steps.

| method | Robin MSE | interior MSE | affine RMSE | time |
|---|---|---|---|---|
| base | $1.95\times10^{-1}$ | $1.16\times10^{-5}$ | $8.52\times10^{-3}$ | – |
| soft $\lambda{=}1$ | $3.43\times10^{-6}$ | $1.31\times10^{-5}$ | $8.19\times10^{-3}$ | $8.34\,\mathrm{s}$ |
| soft $\lambda{=}100$ | $9.09\times10^{-7}$ | $4.10\times10^{-5}$ | $8.67\times10^{-3}$ | $8.24\,\mathrm{s}$ |
| soft $\lambda{=}1000$ | $8.07\times10^{-7}$ | $1.60\times10^{-4}$ | $7.76\times10^{-3}$ | $8.33\,\mathrm{s}$ |
| APRNN/CVXPY-style | $4.99\times10^{-30}$ | $3.06\times10^{-4}$ | $5.66\times10^{-17}$ | $0.14\,\mathrm{s}$ |
| mPOLICE | $9.32\times10^{-30}$ | $9.13\times10^{-5}$ | $7.87\times10^{-17}$ | $6.27\,\mathrm{s}$ |

## E2. DeepONet Operator Learning (smaller-scale variant of Section 4.2)

We train a DeepONet on the same Poisson operator and the same two internal ground pads used in Section 4.2, but with a smaller two-hidden-layer, width-32 trunk of rank 8 to keep the initial APRNN repair and 20 periodic re-projection calls practical. We compare three workflows. *APRNN-freeze* repairs and freezes the trunk before training the branch. *APRNN-reproject* trains both components and re-invokes `VPolytopeRepair` every 5 epochs. *mPOLICE-reproject* uses the corresponding mPOLICE projected-training loop.

Table 3: E2, DeepONet. The left block reports max trunk constraint violation. The right block reports relative $L^2$ test error outside the pads.

| | max trunk violation | | | rel. $L^2$ | | |
|---|---|---|---|---|---|---|
| epoch | APRNN freeze | APRNN reproj. | mPOL reproj. | APRNN freeze | APRNN reproj. | mPOL reproj. |
| 0 | $2.56\times10^{-14}$ | $2.56\times10^{-14}$ | $2.95\times10^{-15}$ | 1.19 | 1.19 | 1.45 |
| 5 | $2.56\times10^{-14}$ | $4.15\times10^{-16}$ | $1.02\times10^{-12}$ | 1.15 | 1.16 | 1.35 |
| 10 | $2.56\times10^{-14}$ | $4.90\times10^{-12}$ | $7.80\times10^{-13}$ | 1.12 | 1.10 | 1.22 |
| 40 | $2.56\times10^{-14}$ | $5.67\times10^{-14}$ | $1.48\times10^{-13}$ | 1.04 | 1.03 | 1.35 |
| 80 | $2.56\times10^{-14}$ | $9.06\times10^{-14}$ | $8.77\times10^{-13}$ | 1.00 | 0.97 | 1.02 |
| 100 | $2.56\times10^{-14}$ | $4.03\times10^{-13}$ | $3.51\times10^{-13}$ | 0.99 | 0.99 | 1.05 |

APRNN-freeze preserves the repaired trunk constraint by keeping the trunk fixed. APRNN-reproject and mPOLICE-reproject re-establish their constraints every 5 epochs during 100 epochs of continued SGD, with interim drift between projections. All three achieve a similar task fit, with relative $L^2$ error 0.99–1.05 at epoch 100. The dense-sample residual uses 200 fixed samples per pad, shared across methods and epochs, and is evaluated after projection. Numerical verification follows every re-projection cycle. The mPOLICE QP configuration passes its initial repair and all 20 cycles with strictly positive margins near $10^{-4}$, pairwise pattern uniqueness, vertex residuals at most $1.1\times10^{-12}$, and finite parameters. The APRNN adapter's stage checks also pass on all 20 cycles. Across all logged epochs, the worst post-projection dense residual is $4.9\times10^{-12}$ for APRNN-reproject and $1.0\times10^{-12}$ for mPOLICE-reproject. On this small trunk, mPOLICE is $4.7\times$ faster per re-projection cycle, taking $0.026\,\mathrm{s}$ instead of $0.121\,\mathrm{s}$. Verification and stage-check evaluation are excluded from both timers. Experiment E8 shows how this gap changes with vertex count.

## E3. Coordinate-Regression Drift

We probe the projected-training step on a coordinate MLP with two Dirichlet pads. mPOLICE re-projects every 5 epochs during 80 epochs of continued task SGD. Without re-projection, both methods drift far from the constraint (max pad violation 0.80 for once-repaired APRNN and 0.92 for once-enforced mPOLICE by

epoch 80), while the re-projected trace stays at numerical precision throughout ($\leq 1.2 \times 10^{-13}$ over all logged epochs). The single repair call costs $0.179\,\text{s}$ for APRNN versus $0.029\,\text{s}$ for mPOLICE.

Table 4: E3, mPOLICE projected-training trace.

| epoch | max pad violation |
|---|---|
| 0 | $2.53 \times 10^{-14}$ |
| 5 | $7.11 \times 10^{-14}$ |
| 10 | $2.09 \times 10^{-14}$ |
| 15 | $2.66 \times 10^{-14}$ |
| 30 | $9.96 \times 10^{-14}$ |
| 50 | $1.82 \times 10^{-14}$ |
| 80 | $5.06 \times 10^{-14}$ |

### E4. Positivity and Boundedness

For qualitative inequalities, we enforce tightened constraints with margin $\eta = 10^{-3}$. A positive signed tightened residual means the *tightened* constraint is missed, while a negative value means slack. APRNN has small positive tightened residuals for positivity and boundedness ($+2.1 \times 10^{-10}$ and $+2.2 \times 10^{-10}$, respectively), while mPOLICE has a small positive tightened positivity residual ($+4.6 \times 10^{-11}$) and a negative tightened boundedness residual ($-1.1 \times 10^{-11}$). Thus, each method misses at least one tightened target at solver scale. The corresponding provable exact-arithmetic statement instead concerns the *original* property. The checked vertices retain signed slack $\eta - 4.6 \times 10^{-11} \approx 10^{-3} > 0$, which the affine argument extends over the region when its hypotheses hold. Table 5 reports signed residuals for both the tightened and original properties.

Table 5: E4, positivity and boundedness with tightened margins.

| method | task MSE | orig pos. viol | orig bound viol | tight pos residual | tight bound residual |
|---|---|---|---|---|---|
| base | $4.19 \times 10^{-2}$ | $1.09 \times 10^{0}$ | $3.90 \times 10^{-1}$ | $+7.42 \times 10^{-1}$ | $+3.94 \times 10^{-1}$ |
| APRNN/CVXPY-style | $2.12 \times 10^{-1}$ | $0$ | $0$ | $+2.1 \times 10^{-10}$ | $+2.2 \times 10^{-10}$ |
| mPOLICE | $1.37 \times 10^{-1}$ | $0$ | $0$ | $+4.6 \times 10^{-11}$ | $-1.1 \times 10^{-11}$ |

### E5. Region and Vertex Scaling

We use a pretrained four-hidden-layer, width-32 ReLU MLP and disjoint Dirichlet patches placed on a grid. We sweep region count $R$ and vertices per region $V$ over three seeds.

Failure begins at different points for the two configured methods. Raw constraint counts do not determine feasibility. For a constant target, the $V$ rows of a region have rank at most $D+1$ under exact affinity. Feasibility therefore depends on the independent rank of the constraint system, the usable output degrees of freedom, and target consistency. With the QP backend, all mPOLICE seeds pass numerical verification through $R=4, V=10$, corresponding to 40 rows with independent rank at most 12. At $R=8$, its numerical checks identify the failed runs. Under the released `eval_3` configuration's $\pm 3$ delta bounds and default centroid references, APRNN begins failing at $R=2$. At $R=2$ and in some $R=8$ runs, the reference patterns are distinct and every Shift succeeds, but the final Assert is infeasible. Other $R=8$ runs fail at the first Shift. The failures therefore occur at multiple stages and are not explained solely by reference-pattern collisions.

### E6. Network Width and Depth

We fix a four-patch nonzero Dirichlet task and sweep MLP width and depth.

Table 6: E5, scaling in regions $R$ and vertices per region $V$. The released-objective APRNN adapter uses $\pm 3$ parameter-delta bounds, and mPOLICE uses the QP backend. "ok" counts seeds (of 3) passing numerical verification. The mPOLICE routine checks strictly positive margins, uniqueness, residuals, and finiteness, while APRNN uses per-stage adapter checks. Residual columns are maximum vertex residuals over passing seeds.

| $R$ | $V$ | total $V$ | APRNN time | ok | mPOLICE time | ok | APRNN res | mPOLICE res |
|---|---|---|---|---|---|---|---|---|
| 1 | 4 | 4 | 0.20 s | 3/3 | 0.008 s | 3/3 | $8\times10^{-15}$ | $9\times10^{-14}$ |
| 1 | 6 | 6 | 0.24 s | 3/3 | 0.008 s | 3/3 | $3\times10^{-15}$ | $1\times10^{-12}$ |
| 1 | 8 | 8 | 0.24 s | 3/3 | 0.009 s | 3/3 | $1\times10^{-14}$ | $2\times10^{-15}$ |
| 1 | 10 | 10 | 0.28 s | 3/3 | 0.009 s | 3/3 | $3\times10^{-14}$ | $5\times10^{-15}$ |
| 2 | 4 | 8 | 0.26 s | 3/3 | 0.010 s | 3/3 | $4\times10^{-14}$ | $1\times10^{-14}$ |
| 2 | 6 | 12 | 0.27 s | 2/3 | 0.012 s | 3/3 | $7\times10^{-15}$ | $1\times10^{-13}$ |
| 2 | 8 | 16 | 0.33 s | 2/3 | 0.014 s | 3/3 | $7\times10^{-15}$ | $2\times10^{-14}$ |
| 2 | 10 | 20 | 0.35 s | 2/3 | 0.016 s | 3/3 | $1\times10^{-14}$ | $2\times10^{-11}$ |
| 4 | 4 | 16 | 0.33 s | 1/3 | 0.127 s | 3/3 | $9\times10^{-15}$ | $2\times10^{-13}$ |
| 4 | 6 | 24 | 0.41 s | 1/3 | 0.132 s | 3/3 | $4\times10^{-14}$ | $5\times10^{-14}$ |
| 4 | 8 | 32 | 0.48 s | 0/3 | 0.136 s | 3/3 | – | $7\times10^{-14}$ |
| 4 | 10 | 40 | 0.56 s | 0/3 | 0.143 s | 3/3 | – | $4\times10^{-13}$ |
| 8 | 4 | 32 | 0.52 s | 0/3 | 0.218 s | 1/3 | – | $8\times10^{-13}$ |
| 8 | 6 | 48 | 0.29 s | 0/3 | 0.244 s | 0/3 | – | – |
| 8 | 8 | 64 | 0.15 s | 0/3 | 0.324 s | 0/3 | – | – |
| 8 | 10 | 80 | 0.17 s | 0/3 | 0.347 s | 0/3 | – | – |

Table 7: E6, network sweep using the released-objective APRNN adapter with the `eval_3` configuration's $\pm 3$ parameter-delta bounds. "ok" counts seeds (of 3) whose run passes the residual check with maximum vertex residual at most $10^{-9}$. These are vertex residuals, not complete numerical verification. APRNN failures are final-Assert infeasibilities under these bounds on this large-amplitude task, with every preceding Shift succeeding. mPOLICE failures are capacity-limited residuals at width 16. MSE columns average the passing seeds.

| width $\times$ depth | APRNN time | ok | mPOLICE time | ok | APRNN outside MSE | mPOLICE outside MSE |
|---|---|---|---|---|---|---|
| $16 \times 2$ | 0.11 s | 0/3 | 0.14 s | 2/3 | – | $2.59\times10^{0}$ |
| $16 \times 3$ | 0.14 s | 0/3 | 0.15 s | 2/3 | – | $1.05\times10^{0}$ |
| $16 \times 4$ | 0.14 s | 0/3 | 0.15 s | 1/3 | – | $5.69\times10^{1}$ |
| $32 \times 2$ | 0.14 s | 0/3 | 0.14 s | 3/3 | – | $5.00\times10^{-1}$ |
| $32 \times 3$ | 0.27 s | 1/3 | 0.16 s | 3/3 | $8.21\times10^{-1}$ | $3.84\times10^{-1}$ |
| $32 \times 4$ | 0.34 s | 1/3 | 0.16 s | 3/3 | $7.36\times10^{-1}$ | $3.93\times10^{-1}$ |
| $64 \times 2$ | 0.46 s | 2/3 | 0.16 s | 3/3 | $5.64\times10^{-1}$ | $2.49\times10^{-1}$ |
| $64 \times 3$ | 0.84 s | 3/3 | 0.19 s | 3/3 | $6.54\times10^{-1}$ | $2.64\times10^{-1}$ |
| $64 \times 4$ | 1.15 s | 3/3 | 0.21 s | 3/3 | $5.61\times10^{-1}$ | $2.39\times10^{-1}$ |
| $128 \times 2$ | 2.49 s | 3/3 | 0.19 s | 3/3 | $6.57\times10^{-1}$ | $1.60\times10^{-1}$ |
| $128 \times 3$ | 4.53 s | 3/3 | 0.24 s | 3/3 | $5.32\times10^{-1}$ | $1.47\times10^{-1}$ |
| $128 \times 4$ | 6.12 s | 3/3 | 0.29 s | 3/3 | $6.33\times10^{-1}$ | $1.42\times10^{-1}$ |

### E7. Piecewise-Affine Dirichlet Boundary

We impose $u(0, y) = g(y) = \sin(\pi y)$ by subdividing the boundary into $N$ strips and matching $g$ at strip vertices. Two design constraints follow from the theory and are respected here. First, the strips are inset by 2% of their height and are therefore *pairwise disjoint*. Distinct sign patterns with a strictly positive margin are infeasible at shared vertices whenever the differing neurons take opposite signs. Only the fragile $\delta = 0$ case permits touching cell closures. Strips sharing complete edges would silently violate the disjointness hypothesis of Theorem 1 for both methods. Second, the width scales with $N$ ($\max(32, 8N)$) so the $4N$ vertex

equality rows stay within the output-layer degrees of freedom. Both methods use the same activation-pin margin ($10^{-6}$). The boundary residual is measured on the union of the covered strips, and the reported floor is the chord-interpolation error of $g$ on the *inset* intervals actually constrained (the correct lower bound for that restricted residual).

Table 8: E7, piecewise-affine Dirichlet with disjoint inset strips (single run, seed 3). Boundary max is measured on the covered strips. The floor is the inset chord-interpolation error. "Fail (detected)" means the method's own checks reported the failure. For mPOLICE this is the hidden-layer sign check before output projection, while for APRNN it is solver failure.

| $N$ | width | APRNN status | APRNN boundary max | mPOLICE status | mPOLICE boundary max | piecewise floor |
|---|---|---|---|---|---|---|
| 1 | 32 | ok | $9.372{\times}10^{-1}$ | ok | $9.372{\times}10^{-1}$ | $9.372{\times}10^{-1}$ |
| 2 | 32 | ok | $1.946{\times}10^{-1}$ | ok | $1.946{\times}10^{-1}$ | $1.946{\times}10^{-1}$ |
| 4 | 32 | ok | $6.49{\times}10^{-2}$ | ok | $6.49{\times}10^{-2}$ | $6.49{\times}10^{-2}$ |
| 8 | 64 | ok | $1.74{\times}10^{-2}$ | fail (detected) | – | $1.74{\times}10^{-2}$ |
| 16 | 128 | fail | – | fail (detected) | – | $4.4{\times}10^{-3}$ |

With inset strips, both methods satisfy the vertex constraints on the covered strips for $N \leq 4$, and the measured boundary error matches the inset chord-interpolation floor to three–four digits. At $N = 8$, APRNN reaches the interpolation floor while this mPOLICE configuration fails its hidden-layer sign check with minimum margin $-1.0$. The check rejects the run before output projection despite 32 equality rows being fewer than the 65 output-layer degrees of freedom. At $N = 16$, APRNN reports solver failure and mPOLICE reports a negative sign margin of $-2.1$. These are configuration-specific failures, not general infeasibility results. The inter-strip gaps cover approximately 4% of the boundary and remain outside the guarantee. Continuity prevents jumps there but gives no quantitative bound. Shared-edge strips would make distinct positive-margin patterns infeasible, which motivates the inset construction. APRNN also permits alternative reference points when centroid patterns are unsuitable (Tao et al., 2023).

**E8. Per-Cycle Re-Projection Cost**

E8 isolates vertex-count scaling for one convex region on a four-hidden-layer, width-32 ReLU MLP. Each method is invoked as one cycle of a projected training loop. APRNN runs full `VPolytopeRepair` with the slice partition of APRNN Algorithm 1. mPOLICE runs sign assignment, sign enforcement, and output projection.

Table 9: E8, stage-wise per-cycle cost (mean over 3 seeds using slice-partition APRNN). "Sign enf." is the hidden-layer sign enforcement only. The mPOLICE total additionally includes sign-pattern assignment ($\approx$1.1–1.5 ms). Timing boundaries are symmetric. Verification is excluded from *both* timers. The adapter's per-stage checks are subtracted, while mPOLICE's full shared verification runs untimed and determines every row's status. All rows pass numerical verification with vertex residuals at numerical precision.

| $V$ | APRNN Shift | APRNN Assert | APRNN total | mPOLICE sign enf. | mPOLICE output | mPOLICE total | ratio |
|---|---|---|---|---|---|---|---|
| 4 | 0.138 s | 0.034 s | 0.173 s | 0.0016 s | 0.0013 s | 0.0040 s | 43× |
| 8 | 0.185 s | 0.035 s | 0.221 s | 0.0019 s | 0.0014 s | 0.0044 s | 50× |
| 16 | 0.252 s | 0.039 s | 0.292 s | 0.0022 s | 0.0012 s | 0.0046 s | 64× |
| 32 | 0.402 s | 0.048 s | 0.452 s | 0.0025 s | 0.0014 s | 0.0051 s | 89× |
| 64 | 0.697 s | 0.077 s | 0.779 s | 0.0030 s | 0.0016 s | 0.0059 s | 132× |
| 128 | 1.467 s | 0.179 s | 1.654 s | 0.0036 s | 0.0022 s | 0.0072 s | 230× |
| 256 | 3.616 s | 0.907 s | 4.539 s | 0.0113 s | 0.0035 s | 0.0162 s | 280× |

The reported E8 timings use APRNN Algorithm 1's slice structure and keep output changes referenced to the original network throughout the stages, following the released `eval_3` objective. The separation follows from solver structure. Each APRNN call solves joint LP/QPs whose size grows with $V$, while mPOLICE's sign enforcement reuses a per-layer factorization across neurons and iterations for a fixed vertex set.

# B    Backend, Trade-off, POLICE, Feasibility, and Dimension Experiments (E9–E13)

This appendix reports five controlled experiments. E9–E11 use the spiral task of Section 4.1, which has two disjoint disks of radius 0.38 centered at $\mp(1.5, 1.05)$ with sign constraints, 32 vertices per disk, and 3 seeds. As in Table 1, the disks are placed where the target field violates the required signs (unconstrained max violation $\approx 9 \times 10^{-2}$), so enforcement is nontrivial. E9 and E11 use three-hidden-layer, width-32 ReLU MLPs. E10 uses the same depth and width in its margin sweep and widths 16–128 in its capacity sweep, while Table 1 itself uses a five-hidden-layer, width-32 network. E12 and E13 use dedicated constructions described below. Algorithm 1 and the configuration tables define the phase templates used by these experiments.

**Spiral-benchmark protocol (Table 1).**   A ReLU MLP with 5 hidden layers of width 32 learns the spiral field on $[-2, 2]^2$. The left disk requires $f \geq 0$, the right requires $f \leq 0$, and the unconstrained field violates these by up to $\approx 8.7 \times 10^{-2}$. Soft penalties sweep $\lambda \in \{1, 10, 100\}$. The benchmark runs in float32 with inequalities tightened by $\eta = 10^{-6}$ to provide a numerical buffer against rounding at the vertices. mPOLICE constrains the inscribed 32-gon of each disk ($\approx 99.4\%$ of the disk area), while all dense metrics are evaluated on the full disk, where measured violations are also zero. The reported metrics are MSE Outside (complement of the constrained disks), MSE Overall, Max Violation, and Affine RMSE (Equation 12, on the disks). MSE Outside is the informative task metric because the enforced cell can extend beyond the target region (Section 2.7), so overall MSE penalizes mPOLICE for affine restrictions on the larger cell. Table 1 reports measured dense-sample quantities and sign margins rather than complete numerical verification (cf. Appendix P).

Tables 10 and 11 summarize every reported configuration, including the paper-scale runs and controlled experiments E1–E13. "Schedules" gives the hidden re-enforcement schedule $S_h$ and the output re-projection schedule $S_o$. *init* means once at initialization. *per. $\kappa$* means every $\kappa$ epochs. *late per.* is periodic but starts only after a configured step. *margin-trig.* fires when the monitored trigger quantity crosses its threshold. *with $S_h$* fires exactly when $S_h$ fires. *final* means once after the training loop, while *off* means never. *one-shot* is a single enforcement invocation with no training loop ($E=0$), and E8 repeats it per timing cycle. The notation $a/b$ gives the cadences of a full-network phase and a subsequent output-only recovery phase. A comma-separated cell lists several firings of the same schedule, such as init and final. Rows with periodic or margin-triggered schedules end with one final enforcement or projection before verification. The 3D-shape and synthetic-field experiments use double precision. Every configuration uses a fixed epoch budget. The RL run additionally uses bounded final enforcement and fine-tuning retries and accepts only a candidate passing its numerical verification and task-performance gates. "Verification" is the numerical-verification mode, run once on the returned network unless marked per cycle. *untimed* means verification runs outside the row's timers. *full* is the shared VERIFY-or-FAIL routine. *full + functional* and *full + flux* add the Robin/Neumann boundary-functional or the wall's dense autograd-flux residual check. *full + dynamics* adds the protected-set and guard geometry, clamp preservation, task-specific output inequalities, deterministic-transition conditions, a dense one-step diagnostic, and checkpoint-reload checks. *task* uses task-specific checks of margins, uniqueness, finiteness, and residuals in E1 and E7. *residual* uses only the vertex-residual check in E6. *sign* is the signed-margin detection studied in E12. *off* means measurements only, without a provable regional guarantee.

## E9. Systematic Backend Comparison Under Identical Settings

All backends receive the same pretrained checkpoint, sign assignment, and margin $\delta = 10^{-3}$ (each solver runs at its native tolerance, $10^{-7}$–$10^{-10}$). In E9–E11, the disk inequalities are tightened by $\eta = 10^{-9}$ to provide a numerical buffer against solver tolerance at the vertices, and shared numerical verification determines every run's status against the original property. Every E9 run passes (for E10's diagnostic $\delta = 0$ rows and E11's deliberately shared-pattern arm, see those subsections). The margin check uses the strict condition $m > 0$. Whether a single bounded solve also *attains* the requested $\delta$ is reported separately wherever it falls short. "POCS→QP" runs a few cheap POCS sweeps and then one QP projection from the POCS iterate (both stages timed). Table 12 confirms the geometric picture of Section 2.6. With a common margin, the returned points from *all* backends have the same verification outcome (min margin $\approx \delta$ and zero dense violation), so constraint satisfaction is equivalent. What differs is the selected feasible point and the wall time. QP returns

---

**Algorithm 1** The mPOLICE training and verification template. Experiments instantiate one or more phases using the settings in Tables 10 and 11. The text below defines the schedules and verification modes, with complexity discussed in Section P.1. Requested-margin acceptance is $m \geq \delta - \epsilon_{\text{solve}}$ and is re-checked after every enforcement. The strict numerical margin criterion is $m > 0$. With $\delta=0$ and verification off, results are measured without a provable regional guarantee.

---

**Require:** pretrained $f_{\boldsymbol{\theta}}$, regions and vertices, stacked constraints from Equation 3, margin $\delta$, and backend $B$
**Require:** training mode, hidden/output schedules $S_h, S_o$ (initial hidden enforcement is unconditional), stopping rule, and budget $E$
**Require:** verification mode $G \in \{\text{full}, \text{task}, \text{residual}, \text{sign}, \text{off}\}$, cadence $\in \{\text{final}, \text{every projection}\}$, and uniqueness mode $U \in \{\text{required}, \text{waived}\}$ (waived only for a declared shared-pattern ablation)
 1: $\{\boldsymbol{S_i}\} \leftarrow \text{ASSIGNSIGNS}(\{R_i\})$ by the mean, majority, or max-margin ECOC rule (§2.5), with identifier bits or near-zero flips aiming at uniqueness when $U = \text{required}$
 2: $\text{ENFORCE}(\boldsymbol{\theta}; \{\boldsymbol{S_i}\}, \delta, B)$ on hidden layers (§2.6)
 3: **if** enforcement is infeasible, required pairwise uniqueness fails, $m$ is nonfinite, or $[m \leq 0 \ \wedge \ (\delta > 0 \ \vee \ G \in \{\text{full}, \text{task}, \text{sign}\})]$ **then**
 4:     repair by re-assignment (§2.5): flip each violating neuron's least-committed region (and, if $U = \text{required}$, resolve duplicate patterns), re-enforce, and after bounded attempts **return** FAIL
 5: **else if** $m < \delta - \epsilon_{\text{solve}}$ **then**
 6:     polish: re-solve the *same* assignment (e.g., a per-neuron QP to tighter solver tolerance), and after bounded retries **return** FAIL (requested margin not met within the retry budget)
 7: **end if**
 8: **if** $S_o \notin \{\text{off}, \text{final}\}$ **then**
 9:     $\text{PROJECTOUTPUT}(\boldsymbol{\theta})$: project final layer onto vertex constraints (§2.9)
10: **end if**
11: **if** verification cadence = every projection **then**
12:     $\text{VERIFY}_G(\boldsymbol{\theta})$. If initial verification fails, **return** FAIL
13: **end if**
14: **for** epoch $= 1, \ldots, E$ **do**
15:     **if** training = penalty **then**
16:         minimize $\mathcal{L}_{\text{task}} + \lambda \mathcal{L}_{\text{con}}$ over mini-batches and increase $\lambda$ on patience expiry (approximate unless $S_o$ projects)
17:     **else**
18:         $T_{\text{SGD}}$ steps of task-only SGD (full network, output layer only, or a configured subset, e.g., branch + trunk output)
19:     **end if**
20:     **if** $S_h$ fires this epoch (margin below $\delta_{\text{trigger}}$, epoch $\equiv 0 \bmod \kappa$, every epoch, or never again for initial-only) **then**
21:         $\text{ENFORCE}(\boldsymbol{\theta}; \{\boldsymbol{S_i}\}, \delta, B)$ on hidden layers and re-check $m$ as above
22:     **end if**
23:     **if** $S_o$ fires this epoch (same rules with the vertex violation as trigger, or coupled to fire with $S_h$, while final/off never fire in-loop) **then**
24:         $\text{PROJECTOUTPUT}(\boldsymbol{\theta})$ with optional tangent-space fine-tuning
25:     **end if**
26:     **if** verification cadence = every projection and $S_h$ or $S_o$ fired **then**
27:         $\text{VERIFY}_G(\boldsymbol{\theta})$. If cycle verification fails, **return** FAIL
28:     **end if**
29:     **if** stopping = early-stop **then**
30:         evaluate $V(\boldsymbol{\theta})$ from Section 2.8 and **break** if $V(\boldsymbol{\theta}) \leq \varepsilon$
31:     **end if**
32: **end for**
33: **if** $S_o = \text{final}$ or the phase ends with a closing enforcement/projection (per experiment, Tables 10 and 11) **then**
34:     final $\text{ENFORCE}$ and/or $\text{PROJECTOUTPUT}$ before verification and re-check $m$ as above
35: **end if**
36: **if** $G \neq \text{off}$ **then**
37:     $\text{VERIFY}_G(\boldsymbol{\theta})$: evaluate the mode-$G$ checks (full includes uniqueness when $U = \text{required}$ and may add functional/flux checks, while other modes follow Table 11)
38:     **if** any check of mode $G$ fails **then**
39:         **return** FAIL (numerical verification failed)
40:     **end if**
41:     **return** $\boldsymbol{\theta}$ with the verification values
42: **else**
43:     **return** $\boldsymbol{\theta}$ with measured residuals (not numerically verified)
44: **end if**

---

Table 10: Algorithm 1 configuration of the paper-scale experiments (schedule and verification vocabulary in the text above).

| Experiment | Assignment | Backend | Margin $\delta$ | Training variant | Schedules $S_h$; $S_o$ | Verification |
|---|---|---|---|---|---|---|
| Spiral disks (§4.1) | mean | POCS | $10^{-3}$ | task-only output-layer fine-tune | init; per. 10 | off |
| DeepONet (§4.2) | max-margin, ECOC | ADMM + QP polish | $10^{-4}$ | geometry-guided preconditioning before enforcement, then branch + trunk-output recovery | init; init, final | full |
| Robin (§4.3) | max-margin, ECOC | ADMM + QP polish | $10^{-4}$ | task-only output-layer recovery (hidden frozen) | init; per. 25 | full + functional |
| Neumann (App. I) | max-margin, ECOC | ADMM + QP polish | $10^{-4}$ | fine-tune, then output-layer recovery | per. 50; per. 50/25 | full + functional |
| RL (§4.4) | max-margin, ECOC | late ADMM + final QP | $10^{-3}$ check $(1.5 \times 10^{-3}$ solve) | TD3 + bounded final enforce–fine-tune retries | late per., final retry; final retry | full + dynamics, final/reload |
| Performance (§4.5) | geometry-feasible one-vs-rest | QP, POCS, ADMM | $10^{-3}$ | single invocation | one-shot | full, timed |
| Synthetic field (App. M) | max-margin, ECOC | ADMM + QP polish | $10^{-4}$ | projected training, then output-layer recovery | margin-trig./off; with $S_h$/per. 1 | full |
| Impermeable wall (App. J) | mean (dead neurons harmonized) | ADMM + QP polish | $10^{-4}$ | fine-tune, then output-layer recovery | per. 50; per. 50/20 | full + flux |
| 3D shape (App. K) | mean | QP | $10^{-4}$ check $(1.2 \times 10^{-4}$ solve) | geometry-guided filled-body preconditioner, then rank-reduced constrained ridge LS | final; final | full |

each neuron's minimal-change projection to solver tolerance, with layers processed sequentially rather than as a global minimal-norm repair. POCS matches its movement norm to two decimals, while ADMM moves parameters $\approx 1.9\times$ more. E9–E11 use a direct output-layer QP followed by the common vertex-residual check. Table 13 isolates enforcement wall time as the vertex count grows. POCS/ADMM stay in the millisecond range through 2048 total vertices, while the per-neuron QP grows superlinearly, from $0.010\,\text{s}$ at $V{=}16$ per disk to $5.6\,\text{s}$ at $V{=}1024$. This sweep times one bounded solve per cell and reports the margin that solve attains. The QP and POCS entries land at $\approx \delta$, while ADMM's iteration-capped solves scatter around it (from $8.9 \times 10^{-4}$ at $V{=}64$ to $1.5 \times 10^{-3}$ at $V{=}16$). The $V{=}64$ entry passes the strict positive-margin check yet misses the requested margin, a shortfall that Algorithm 1's acceptance rule would send to the polish step (and FAIL if persistent) rather than accept.

### E10. Quantifying the Loss vs. Activation-Cell-Size Trade-off

For each run we estimate the area of the activation cell containing each disk by Monte Carlo pattern matching using $4 \times 10^5$ samples over the $[-2, 2]^2$ domain, where the disk area is 0.454. Table 14 shows both sweeps. In this three-seed configuration, margins $\delta \in [10^{-3}, 10^{-2}]$ give cell/disk ratios 1.2–1.3 with little measured task-error change. This is an empirical range, not a cost-free general rule. At $\delta = 0$, two seeds have slightly negative final margins ($-4.6 \times 10^{-8}$ and $-2.1 \times 10^{-6}$), so the strict positive-margin check fails, whereas every $\delta \geq 10^{-3}$ run passes it. Larger margins coincide with larger cells and errors. From width 16 to 128, the sampled cell/region ratio falls from 1.44 to 1.04 and the constrained/unconstrained MSE ratio from 7.8 to 2.2. These results do not establish a monotonic relationship. Max sampled violation is 0 in every passing row. The margin sweep tracks requested $\delta$ within 0.5%, while five of nine capacity-sweep solves stop at

Table 11: Algorithm 1 configuration of the controlled experiments E1–E13 (schedule and verification vocabulary in the text above).

| Experiment | Assignment | Backend | Margin $\delta$ | Training variant | Schedules $S_h$; $S_o$ | Verification |
|---|---|---|---|---|---|---|
| E1 Robin head-to-head | max-margin, ECOC | ADMM | $10^{-4}$ | task-only output-layer fine-tune | init; per. 10 | task |
| E2 DeepONet drift | max-margin, ECOC | QP | $10^{-4}$ | re-projection loop | per. 5; per. 5 | full, per cycle |
| E3 coordinate drift | max-margin, ECOC | ADMM | $10^{-4}$ | re-projection loop | per. 5; per. 5 | off |
| E4 positivity/boundedness | max-margin, ECOC | QP | $10^{-4}$ | task-only output-layer fine-tune | init; per. 10 | full |
| E5 region/vertex scaling | max-margin, ECOC | QP | $10^{-4}$ | single invocation | one-shot | full |
| E6 network sweep | max-margin, ECOC | ADMM | $10^{-4}$ | single invocation | one-shot | residual |
| E7 piecewise Dirichlet | max-margin, ECOC | QP | $10^{-6}$ | single invocation | one-shot | task |
| E8 vertex scaling | max-margin, ECOC | ADMM | $10^{-4}$ | single invocation per cycle | one-shot per cycle | full, untimed, per cycle |
| E9 backend comparison | mean | as stated | $10^{-3}$ | task-only output-layer fine-tune | init; per. 10 | full |
| E10 margin/width sweeps | mean | ADMM | sweep {0, $10^{-3}, \ldots,$ 0.2} | task-only output-layer fine-tune; widths 16–128 | init; per. 10 | full ($\delta=0$ rows diagnostic) |
| E11 POLICE-style ablation | mean | QP | $10^{-3}$ | task-only output-layer fine-tune | init; per. 10 | full (uniqueness waived by design, shared-pattern arm) |
| E12 infeasibility repair | mean + adversarial | QP | $10^{-3}$ | enforcement only | init; off | sign |
| E13 dimension scaling | mean | ADMM | $10^{-4}$ | enforcement + projection | one-shot | full |

Table 12: E9, backends under identical $\delta$ and sign assignment, using solver-native tolerances (mean over 3 seeds). Enforce time is the hidden-layer projection only. The unconstrained reference has MSE outside $2.1\times10^{-3}$ and maximum violation $9.3\times10^{-2}$.

| backend | enforce time | $\|\Delta\theta\|$ | min margin | MSE outside | max viol. | affine RMSE |
|---|---|---|---|---|---|---|
| QP | $0.021\,\text{s}$ | 0.47 | $1.0\times10^{-3}$ | $7.45\times10^{-3}$ | 0 | $2.9\times10^{-15}$ |
| POCS | $0.008\,\text{s}$ | 0.47 | $1.0\times10^{-3}$ | $7.41\times10^{-3}$ | 0 | $2.8\times10^{-15}$ |
| ADMM | $0.042\,\text{s}$ | 0.89 | $1.0\times10^{-3}$ | $9.60\times10^{-3}$ | 0 | $2.6\times10^{-15}$ |
| POCS→QP | $0.022\,\text{s}$ | 0.47 | $1.0\times10^{-3}$ | $7.41\times10^{-3}$ | 0 | $2.8\times10^{-15}$ |

$7.8\times10^{-4}$–$9.9\times10^{-4}$ instead of $10^{-3}$. Algorithm 1 would polish or fail those shortfalls. Their strict $m > 0$ checks support only the conditional affinity statement, not attainment of the requested margin.

Table 13: E9, enforcement wall time vs. vertices per disk (seed 0). Margins are those a single bounded solve attains. ADMM's $V{=}64$ solve stops below the requested $\delta$ (see text). The per-neuron QP grows superlinearly.

| | QP | | POCS | | ADMM | |
|---|---|---|---|---|---|---|
| $V$/disk | time | margin | time | margin | time | margin |
| 16 | 0.010 s | $1.0{\times}10^{-3}$ | 0.008 s | $1.0{\times}10^{-3}$ | 0.004 s | $1.5{\times}10^{-3}$ |
| 64 | 0.036 s | $1.0{\times}10^{-3}$ | 0.008 s | $1.0{\times}10^{-3}$ | 0.049 s | $8.9{\times}10^{-4}$ |
| 256 | 0.261 s | $1.0{\times}10^{-3}$ | 0.009 s | $1.0{\times}10^{-3}$ | 0.013 s | $1.3{\times}10^{-3}$ |
| 1024 | 5.620 s | $1.0{\times}10^{-3}$ | 0.014 s | $1.0{\times}10^{-3}$ | 0.011 s | $1.0{\times}10^{-3}$ |

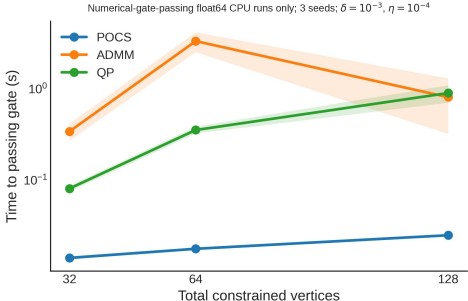

Figure A1: mPOLICE overhead vs. total constrained vertex count, from the performance benchmark of Section 4.5 (setup in Appendix N). This cost is training-time only.

### E11. Single-Pattern POLICE-Style Ablation vs. mPOLICE

We isolate the shared-pattern convex-hull effect by copying the left disk's sign pattern onto the right disk, so successful enforcement places both disks in one activation cell and the network is provably affine on their convex hull in exact arithmetic. This controlled ablation differs from POLICE by updating both weights and biases and applying output projection. Only the assigned pattern is varied. Table 15 reports one shared cell of sampled area $\approx 3.4$ ($\approx 7.6\times$ the disk area), central-hull affine RMSE $1.5{\times}10^{-15}$ (versus 0.59 unconstrained), and $19\times$ higher outside MSE than mPOLICE. mPOLICE's separate sampled cells have area $\approx 0.54$ ($1.19\times$ the disk) and central-hull affine RMSE 0.55. The two affine-RMSE columns have different interpretations: low disk affine RMSE is desired because the constrained disks should be affine, whereas near-zero central-hull affine RMSE is the *undesired* signature of shared-pattern coupling across the unconstrained gap. Among constraint-satisfying rows, a larger value in the final column is therefore desired in this diagnostic; mPOLICE's 0.55, close to the unconstrained model's 0.59, shows that it does not force the intervening hull to share one affine map. The Delaunay-masked probe covers the polygonal convex hull. An unmasked box mixes unconstrained points and reads $3.6{\times}10^{-2}$ for POLICE-hull. Both rows meet their vertex constraints to the reported numerical tolerance. POLICE-hull deliberately waives pattern uniqueness.

### E12. Sign-Assignment Infeasibility, Detection, and Repair

Three collinear boxes $A, B, C$ receive constraints on a two-hidden-layer, width-16 ReLU MLP. The data-driven mean assignment is feasible. After QP enforcement the signed-margin check passes at $+1.0{\times}10^{-3}$ ($= \delta$). We then adversarially overwrite one first-layer neuron with the infeasible pattern $(+, -, +)$ across $A, B, C$ (no single hyperplane can realize it). The per-neuron projection is infeasible, and the check *fails* at $-2.05$, localizing the failure to exactly that neuron. The bounded repair loop of Section 2.5 flips the sign of the least-committed region for that neuron and verifies that all global patterns remain unique. It then re-enforces the assignment, and the check passes again at $+1.0{\times}10^{-3}$ in one attempt. The same post-hoc logic applies to the constraint-enforcement stage, where vertex residuals provide the check, and it also covers solvers that terminate early with an inaccurate iterate.

Table 14: E10 (mean over 3 seeds). The left block gives the margin sweep at width 32. The $^\dagger$ row at $\delta = 0$ is diagnostic because two of three seeds fail the strict positive-margin check within solver tolerance. The summary statistics omit this row. The right block gives the capacity sweep at $\delta = 10^{-3}$.

| $\delta$ | cell/region | MSE out | width | cell/region | MSE out | base MSE | penalty |
|---|---|---|---|---|---|---|---|
| $0^\dagger$ | 1.20 | $9.58\times10^{-3}$ | 16 | 1.44 | $2.80\times10^{-2}$ | $3.58\times10^{-3}$ | $7.8\times$ |
| $10^{-3}$ | 1.21 | $9.60\times10^{-3}$ | 32 | 1.21 | $9.60\times10^{-3}$ | $2.09\times10^{-3}$ | $4.6\times$ |
| $10^{-2}$ | 1.30 | $1.01\times10^{-2}$ | 64 | 1.11 | $4.78\times10^{-3}$ | $1.44\times10^{-3}$ | $3.3\times$ |
| $5\times10^{-2}$ | 1.52 | $1.05\times10^{-2}$ | 128 | 1.04 | $2.54\times10^{-3}$ | $1.16\times10^{-3}$ | $2.2\times$ |
| $10^{-1}$ | 1.84 | $1.44\times10^{-2}$ | | | | | |
| $2\times10^{-1}$ | 2.30 | $1.62\times10^{-2}$ | | | | | |

Table 15: E11 (mean over 3 seeds). "Central hull" is a 0.8×0.8 box centered between the disks, masked to the convex hull of the two polygons. In the final column, higher is desired among constraint-satisfying methods because it indicates less unintended convex-hull coupling; this direction is specific to the decoupling diagnostic.

| method | MSE outside ↓ | max viol. ↓ | cell area ↓ | disk affine RMSE ↓ | central-hull affine RMSE ↑ |
|---|---|---|---|---|---|
| unconstrained | $2.09\times10^{-3}$ | $9.3\times10^{-2}$ | – | $3.6\times10^{-2}$ | 0.59 |
| POLICE-style (shared pattern) | $1.42\times10^{-1}$ | 0 | 3.44 | $1.7\times10^{-15}$ | $1.5\times10^{-15}$ |
| mPOLICE | $7.45\times10^{-3}$ | 0 | 0.54 | $2.9\times10^{-15}$ | 0.55 |

### E13. Input-Dimension Scaling and the Output-Layer DoF Bound

We sweep the ambient input dimension $D \in \{2, 4, 8, 16, 32, 64\}$ on a three-hidden-layer, width-64 ReLU MLP with two randomly oriented $k$-dimensional simplex regions carrying Dirichlet targets ±0.5. Here $k = \min(D, 4)$, giving a fixed-intrinsic-dimension, low-vertex-count stress test, while the boundary strips and pads used in practice are full-dimensional but thin. Table 16 shows that the total cost of one mPOLICE invocation stays at 6–12 ms and the vertex and dense in-region residuals stay at numerical precision across the entire sweep. Because mPOLICE prescribes cells rather than enumerating them, no cell enumeration occurs. The only potentially exponential quantity is the user-chosen vertex count of full-dimensional V-represented regions (cf. Section P.1). A second block probes *full*-dimensional regions ($k = D$, hence $2(D+1)$ equality constraints). At $D = 8$ the projection succeeds at precision. At $D = 32$, numerical verification flags vertex residuals of $6.0\times10^{-2}$–$4.6\times10^{-1}$. For region $i$, let $\mathbf{\Phi}_i$ contain the penultimate-layer feature rows at its vertices. In two of the three $D = 32$ seeds, each augmented matrix $[\mathbf{\Phi}_i, \mathbf{1}]$ has full rank 33, yet the joint equality system is inconsistent (least-squares residuals $1.5\times10^{-2}$–$1.6\times10^{-2}$). In the third, the two matrices are additionally rank-deficient (ranks 31 and 22) with joint residual $2.0\times10^{-3}$. Thus no output layer can satisfy both regions' targets in these runs. Feasibility is governed by augmented-matrix rank and target consistency. A target constant across all regions can be carried by the bias alone, whereas the opposite ±0.5 targets here cannot, so raw active-unit counts or parameter counts do not determine it (cf. the E5 discussion). This is consistent with the capacity discussion in Section 5.

Table 16: E13, ambient-dimension sweep over 3 seeds. Times are seed means, residual columns are seed maxima, and the regions are $k = \min(D, 4)$-dimensional simplices.

| $D$ | vertices/region | total time | max vertex residual | max dense residual | affine RMSE |
|---|---|---|---|---|---|
| 2 | 3 | 12 ms | $8.9\times10^{-15}$ | $1.9\times10^{-14}$ | $3.3\times10^{-15}$ |
| 4 | 5 | 11 ms | $4.7\times10^{-15}$ | $8.0\times10^{-15}$ | $1.5\times10^{-15}$ |
| 8 | 5 | 9 ms | $4.9\times10^{-15}$ | $8.0\times10^{-15}$ | $1.2\times10^{-15}$ |
| 16 | 5 | 7 ms | $5.8\times10^{-15}$ | $8.0\times10^{-15}$ | $1.1\times10^{-15}$ |
| 32 | 5 | 6 ms | $4.4\times10^{-15}$ | $5.3\times10^{-15}$ | $8.8\times10^{-16}$ |
| 64 | 5 | 6 ms | $3.1\times10^{-14}$ | $2.7\times10^{-14}$ | $8.1\times10^{-16}$ |

## C   Proofs of Theorems 1 and 2

*Proof.* **Part 1. Single Region Affinity.** Fix $i$ and write $R = R_i = \mathrm{conv}\{\boldsymbol{v}_1, \ldots, \boldsymbol{v}_P\}$. We proceed by induction on the layer index $\ell$. The key observation, used at every layer, is that an affine function on a convex set whose vertex values share a common sign maintains that sign throughout the set (any interior point is a convex combination of vertices, so the function value is the same convex combination of same-sign vertex values).

*Base case* ($\ell{=}1$). The identity map is trivially affine on $R$, so each pre-activation $z_k^{(1)}(\boldsymbol{x}) = \boldsymbol{w}_k^{(1)\top}\boldsymbol{x} + b_k^{(1)}$ is affine on $R$. If $z_k^{(1)}(\boldsymbol{v}_p)$ has a consistent sign across all vertices, the observation above implies $z_k^{(1)}$ keeps that sign on all of $R$. Hence $\sigma(z_k^{(1)})$ is either $z_k^{(1)}$ or 0 throughout $R$, and the layer output $\boldsymbol{x}^{(2)}(\boldsymbol{x})$ is affine in $\boldsymbol{x}$ on $R$.

*Inductive step.* Assume $\boldsymbol{x}^{(\ell)}(\boldsymbol{x})$ is affine in $\boldsymbol{x}$ on $R$. Then $z_k^{(\ell)}(\boldsymbol{x}) = \boldsymbol{w}_k^{(\ell)\top}\boldsymbol{x}^{(\ell)}(\boldsymbol{x}) + b_k^{(\ell)}$ is also affine on $R$. The same sign-consistency argument gives that $\sigma(z_k^{(\ell)})$ is affine on $R$, so $\boldsymbol{x}^{(\ell+1)}$ is affine on $R$.

After the $L-1$ hidden layers, composition with the affine output layer gives $f_{\boldsymbol{\theta}}(\boldsymbol{x}) = \boldsymbol{\Lambda}_R\boldsymbol{x} + \boldsymbol{\gamma}_R$ for $\boldsymbol{x} \in R$. Equivalently, $R$ lies in the closed activation-pattern region associated with the prescribed weak signs. Zeros may make the symbolic pattern nonunique without affecting affinity.

**Part 2. Multiple Disjoint Regions with Unique Patterns.** By Part 1, consistent enforcement of pattern $\boldsymbol{S}_i$ across the vertices of $R_i$ places $R_i$ inside the associated closed activation-pattern region $\mathcal{P}_i$. Distinctness is where the strict-margin hypothesis enters. Fix $i \neq j$ and let $z$ be the pre-activation of a neuron on which $\boldsymbol{S}_i$ and $\boldsymbol{S}_j$ differ and which attains a strictly positive margin $m > 0$ on both regions. By the argument of Part 1, $z$ is affine on each region, and an affine function whose values at every vertex of a convex hull are $\geq m$ (resp. $\leq -m$) keeps that strict sign at every convex combination. After orienting signs, $z > 0$ throughout $R_i$ and $z < 0$ throughout $R_j$. A closed activation-pattern region fixes one weak sign of $z$, so no such region contains both $R_i$ and $R_j$. In particular $\mathcal{P}_i \neq \mathcal{P}_j$. Hence the two affine restrictions are not constrained to coincide, and no shared-pattern constraint forces a common affine map over their convex hull. □

*Proof of Theorem 2.* For any $\boldsymbol{x} \in R_i$, choose convex coefficients $\alpha_p \geq 0$ with $\sum_p \alpha_p = 1$ and $\boldsymbol{x} = \sum_p \alpha_p \boldsymbol{v}_p^{(i)}$. The residual is affine, hence equals the same convex combination of its vertex residuals. Therefore

$$\|\boldsymbol{E}f_{\boldsymbol{\theta}}(\boldsymbol{x}) - \boldsymbol{g}_i(\boldsymbol{x})\| \leq \sum_p \alpha_p \left\|\boldsymbol{E}f_{\boldsymbol{\theta}}(\boldsymbol{v}_p^{(i)}) - \boldsymbol{g}_i(\boldsymbol{v}_p^{(i)})\right\|,$$

which is bounded by the largest vertex residual. Applying the same convex-combination identity componentwise proves the inequality statement, and restriction to a face preserves affinity. □

## D   Constraint Reductions on Planar Strips

Once affinity holds on $R_i$, any constraint linear in $(\boldsymbol{\Lambda}_i, \boldsymbol{\gamma}_i)$ and identifiable from its vertex values reduces to Equation 3. Dirichlet data select rows of $\boldsymbol{Y}_i$, while interface continuity or jump conditions couple corresponding rows from incident region blocks.

For a codimension-one boundary, a zero-thickness face does not identify the ambient normal slope, so we thicken it into a full-dimensional strip $R^\varepsilon$. For the wall $x = 0$ with interior $x > 0$, outward normal $\boldsymbol{n} = -\boldsymbol{e}_x$, and paired vertices $(0, y_j)$ and $(\varepsilon, y_j)$, let $\partial_n f$ denote the interior one-sided gradient trace. A positive-margin pattern also makes it the ordinary derivative in a neighborhood of the face. Affinity gives

$$\partial_n f(0, y_j) = -\partial_x f(0, y_j) = \frac{f(0, y_j) - f(\varepsilon, y_j)}{\varepsilon}. \tag{14}$$

Thus Neumann data $\partial_n f = q_j$ become

$$f(0, y_j) - f(\varepsilon, y_j) = \varepsilon q_j, \tag{15}$$

and the Robin law becomes

$$hf(0, y_j) + \frac{k}{\varepsilon}\big(f(0, y_j) - f(\varepsilon, y_j)\big) = hT_{\mathrm{amb}}(y_j). \tag{16}$$

For constant coefficients and boundary data affine along the face, these residuals are affine in the face coordinates, so Theorem 2 extends the maximum vertex-residual bound over the face. Non-affine data require a compatible subdivision or a separately bounded approximation error.

If the two paired output values are each perturbed by at most $\xi$, the induced normal-derivative perturbation can be as large as $2\xi/\varepsilon$, and the Robin perturbation as large as $|h|\xi + 2|k|\xi/\varepsilon$. The derivative experiments therefore check the assembled boundary-functional residual directly rather than taking $\varepsilon$ arbitrarily small.

## E    Bias-Only Contradiction Example

Adjusting only the bias suffices for a single region (as in POLICE) but can fail for multiple disjoint regions with conflicting sign requirements.

Consider a neuron with pre-activation $z(x) = wx + b$, fixed $w > 0$, and two regions $R_1 = [0, 1]$, $R_2 = [2, 3]$. We require $z(x) \geq 0$ on $R_1$ and $z(x) \leq 0$ on $R_2$. Since $z$ is increasing, the first condition gives $b \geq 0$ (evaluating at $x = 0$) and the second gives $b \leq -3w$ (evaluating at $x = 3$). Because $w > 0$, no $b$ can satisfy both inequalities simultaneously.

This contradiction shows that bias-only adjustments lack the flexibility to enforce independent sign patterns across disjoint regions, motivating mPOLICE's joint weight-and-bias enforcement.

## F    Non-Convex Approximation

mPOLICE requires each constrained region to be a convex polytope. A non-convex region can be approximated by decomposing it into disjoint convex sub-regions $\{\mathcal{R}_j\}$ and enforcing affinity on each independently. The provable exact-arithmetic result holds within a sub-region only when its sign assignment is feasible with positive realized margins and its vertex system is compatible. The approximation quality also depends on how tightly the sub-regions tile the target shape.

Figure A2 illustrates this on a saddle field with two square affine regions. When the squares are far apart (left), the gap spans several activation cells with no affine guarantee. When they are placed with an arbitrarily small positive gap (right), the uncovered sliver between them becomes negligible in measure and the pair approximates one non-convex affine region. The gap does not mathematically disappear, because behavior inside it is continuous but carries no provable quantitative bound. Shrinking it toward zero eventually makes strictly positive-margin sign constraints for opposite patterns numerically incompatible.

## G    Reinforcement Learning Problem Details

This single-seed (seed 40) control example provides an exact-arithmetic collision-avoidance guarantee for the returned deterministic actor under the stated kinematic model. Ordinary float32 checks are repeated after checkpoint serialization and reload. The guarantee does not cover finite-precision execution or training-time exploration noise.

We summarize the setup and hyperparameters below. The accompanying code provides the exact implementation details.

The reinforcement learning environment is a continuous 2D navigation task within the normalized square domain $[-1, 1] \times [-1, 1]$. The agent navigates from a starting position to a target location while avoiding two horizontal line obstacles. The agent is trained using the Twin Delayed Deep Deterministic Policy Gradient (TD3) algorithm (Fujimoto et al., 2018).

**State Space.**    The state $s \in \mathbb{R}^2$ represents the agent's current $(x, y)$ coordinates within the normalized environment.

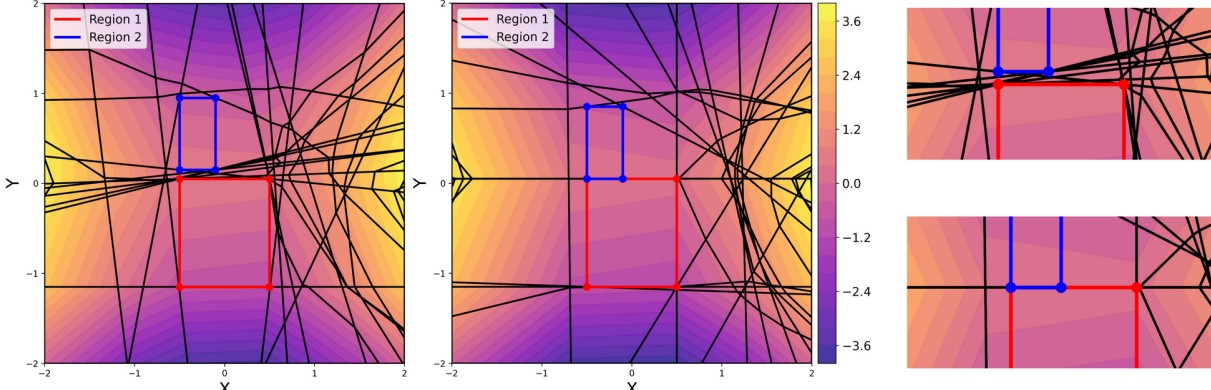

Figure A2: Two affine regions approximating a saddle field. A large gap spans several polytopes with no affine guarantees *(left)*. Close placement shrinks the uncovered gap to negligible measure, approximating a non-convex shape *(right)*. The gap itself remains continuous but carries no provable quantitative bound.

**Action Space.** Let $g_{\theta}(s) \in \mathbb{R}^2$ denote the raw actor output. The deployed deterministic action is $a_{\theta}(s) = \text{clip}(g_{\theta}(s), [-1, 1]^2)$. The unclipped state update is $s^+ = s + \Delta t\, a_{\theta}(s)$ with $\Delta t = 0.2$, so each coordinate can move by at most $q = 0.2$ in one step. An out-of-domain update terminates the episode.

**Environment Features.**

- **Target**: A fixed point $s_T = (0.8, 0.8)$.

- **Obstacles**: Two horizontal line segments. For $k \in \{1, 2\}$, obstacle $k$ is defined by its y-coordinate $y_{O_k}$ and an x-interval $[x_{O_k}^{\min}, x_{O_k}^{\max}]$. The parameters used are:

    - Obstacle 1: $y_{O_1} = 0.3$, $x \in [-0.3, 0.1]$.
    - Obstacle 2: $y_{O_2} = -0.3$, $x \in [0.55, 0.95]$.

**Protected sets and predecessor guards.** Write obstacle $k$ as $O_k = [\ell_k, r_k] \times \{c_k\}$ and set the clearance to $\epsilon_c = 0.01$. Its closed protected rectangle is

$$\mathcal{U}_k = [\ell_k - \epsilon_c, r_k + \epsilon_c] \times [c_k - \epsilon_c, c_k + \epsilon_c]. \tag{17}$$

Let $\underline{x}_k, \overline{x}_k, \underline{y}_k, \overline{y}_k$ denote the four corresponding endpoints. The one-step predecessor guard is

$$\mathcal{G}_k = \left(\mathcal{U}_k \oplus [-q, q]^2\right) \cap [-1, 1]^2. \tag{18}$$

Thus its radius relative to the original line obstacle is $q + \epsilon_c = 0.21$. The two guards are

$$\mathcal{G}_1 = [-0.51, 0.31] \times [0.09, 0.51], \qquad \mathcal{G}_2 = [0.34, 1] \times [-0.51, -0.09].$$

The guards enter the proof but are omitted from Figure 4 for clarity. The red regions show the much thinner protected sets $\mathcal{U}_k$.

The set $\mathcal{G}_k \setminus \mathcal{U}_k$ is covered by four convex rectangular slabs below, above, left, and right of $\mathcal{U}_k$. With distance-retention factor $\rho = 0.05$ and output margin $\eta > 0$, the raw actor is constrained at every vertex of

the respective slab by

$$g_{\boldsymbol{\theta},y}(\boldsymbol{s}) \leq \frac{1-\rho}{\Delta t}(\underline{y}_k - s_y) - \eta \qquad \text{below,} \qquad (19)$$

$$g_{\boldsymbol{\theta},y}(\boldsymbol{s}) \geq \frac{1-\rho}{\Delta t}(\overline{y}_k - s_y) + \eta \qquad \text{above,} \qquad (20)$$

$$g_{\boldsymbol{\theta},x}(\boldsymbol{s}) \leq \frac{1-\rho}{\Delta t}(\underline{x}_k - s_x) - \eta \qquad \text{left,} \qquad (21)$$

$$g_{\boldsymbol{\theta},x}(\boldsymbol{s}) \geq \frac{1-\rho}{\Delta t}(\overline{x}_k - s_x) + \eta \qquad \text{right.} \qquad (22)$$

The final output QP solves these inequalities with $\eta = 2 \times 10^{-3}$. The acceptance check requires $\eta = 10^{-3}$ with tolerance $10^{-6}$. Positive-margin hidden-layer enforcement makes $g_{\boldsymbol{\theta}}$ affine on each complete guard $\mathcal{G}_k$. Each displayed residual is therefore affine, so its vertex inequalities extend to the whole slab. The numerical gate also checks that every upper-bound threshold is at least $-1$ and every lower-bound threshold is at most 1, which is exactly the condition needed for componentwise clipping to preserve the corresponding one-sided inequality.

**Collision-avoidance implication.** For any $\boldsymbol{s} \in \mathcal{G}_k \setminus \mathcal{U}_k$, choose a slab containing $\boldsymbol{s}$, let $d(\boldsymbol{s}) > 0$ be its outward signed distance from the corresponding face of $\mathcal{U}_k$, and let $\sigma \in \{-1, 1\}$ be that outward coordinate direction. The four slab constraints, after action clipping, share the form

$$\sigma a_{\boldsymbol{\theta},j}(\boldsymbol{s}) \geq -\frac{1-\rho}{\Delta t} d(\boldsymbol{s}) + \eta.$$

Along the straight transition segment $\boldsymbol{p}(\lambda) = \boldsymbol{s} + \lambda \Delta t\, \boldsymbol{a}_{\boldsymbol{\theta}}(\boldsymbol{s})$, $0 \leq \lambda \leq 1$, its signed distance satisfies

$$d(\boldsymbol{p}(\lambda)) \geq \big(1 - \lambda(1-\rho)\big)d(\boldsymbol{s}) + \lambda \Delta t\, \eta > 0. \qquad (23)$$

The entire segment therefore remains on the same side of $\mathcal{U}_k$. If $\boldsymbol{s} \notin \mathcal{G}_k$, the componentwise step bound $q = 0.2$ prevents the segment from reaching $\mathcal{U}_k$. Applying the argument to both obstacles and inducting over nonterminal in-domain states proves that every deterministic trajectory initialized in $[-1,1]^2 \setminus \bigcup_k \mathcal{U}_k$ avoids both protected rectangles, and hence both line obstacles, until target attainment or out-of-bounds termination.

After serialization and reload, the returned float32 checkpoint has minimum hidden signed margin $1.499731 \times 10^{-3}$ against the requested $10^{-3}$ check, complete and distinct patterns, finite parameters, and minimum raw output-inequality slack $1.999855 \times 10^{-3}$ (equivalently, $9.99855 \times 10^{-4}$ beyond the required output margin). Guard geometry and clamp preservation also pass. A $161 \times 161$ one-step grid diagnostic finds zero obstacle intersections and zero next-state entries into the protected rectangles over 25,855 admissible states. This sampled diagnostic is corroborating evidence, not the source of the region-wide result. On the same 100 fixed admissible evaluation starts, the deterministic policy obtains mean reward 4.507979 with zero observed collisions. Because this result uses one training seed, it does not establish robustness across seeds.

**Training buffers.** The 0.1-high rectangle directly below each obstacle is denoted $\mathcal{B}_k$ and is used only for training-state sampling, reward bookkeeping, and late training-time affinity steps. It is distinct from both the protected set $\mathcal{U}_k$ and the full predecessor guard $\mathcal{G}_k$. Its vertices are

$$(\ell_k, c_k - 0.1),\ (r_k, c_k - 0.1),\ (r_k, c_k),\ (\ell_k, c_k).$$

**Reward Function.** Let $\boldsymbol{s}_{t+1}^{\text{raw}} = \boldsymbol{s}_t + \Delta t\, \boldsymbol{a}_t$ and $\tilde{\boldsymbol{s}}_{t+1} = \text{clip}(\boldsymbol{s}_{t+1}^{\text{raw}}, [-1,1]^2)$. Define $d_{t+1} = \|\tilde{\boldsymbol{s}}_{t+1} - \boldsymbol{s}_T\|_2$ and $b_{t+1} = \mathbb{I}(d_{t+1} < 0.1\ \wedge\ \tilde{\boldsymbol{s}}_{t+1} \notin \bigcup_k \mathcal{B}_k)$. The implemented reward is piecewise:

$$r_t = \begin{cases} -20, & \text{if the transition segment intersects an obstacle,} \\ -10, & \text{if there is no collision but } \boldsymbol{s}_{t+1}^{\text{raw}} \notin [-1,1]^2, \\ -d_{t+1} + 10b_{t+1}, & \text{otherwise.} \end{cases}$$

Thus collision and out-of-bounds transitions receive only their terminal penalty. Distance shaping and the target bonus apply only otherwise. The components are:

- $-\|\tilde{s}_{t+1} - s_T\|_2$ is the nonterminal distance-shaping term.

- $R_{\text{target}} = +10$ is added when the clipped next state lies within 0.1 of $s_T$ and outside every training buffer $\mathcal{B}_k$.

- $P_{\text{collision}} = -20$ is incurred if the agent's path segment from $s_t$ to $s_{t+1}$ intersects an obstacle.

- $P_{\text{bounds}} = -10$ is incurred if the agent moves outside the environment boundaries.

The episode terminates upon reaching the target, collision, or out-of-bounds transition. Reaching 500 steps is a time-limit truncation and is stored with the replay-buffer terminal flag set to zero.

**Training and returned-policy selection.** TD3 runs for 20,000 steps. Periodic ADMM affinity steps on the training buffers $\mathcal{B}_k$ begin at step 16,000 and use zero training margin. The bounded final loop assigns max-margin ECOC patterns on the full guards $\mathcal{G}_k$, applies a per-neuron hidden QP at solve margin $1.5 \times 10^{-3}$, and projects the four slab constraints by an output QP. A candidate is returned only after the float32 numerical verification gate, checkpoint-reload check, zero-collision evaluation gate, and mean-reward threshold 4.5 all pass. If any check fails, TD3 fine-tuning and verification repeat for at most 50 iterations. This final protocol preserves the actor architecture and adds no deployed inference module.

The hyperparameters used in the reinforcement learning experiment (Section 4.4) are detailed in Tables 17 and 18.

Table 17: Environment, sampling, and TD3 hyperparameters for the reinforcement learning experiment.

| Category | Parameter | Value |
|---|---|---|
| Environment | State dimensionality | 2 |
| | Action dimensionality | 2 |
| | Max action-component magnitude | 1.0 |
| | Time step ($\Delta t$) | 0.2 |
| | Max episode length | 500 |
| | Training-sampling buffer height | 0.1 |
| | Protected clearance | 0.01 |
| | Predecessor-guard radius | 0.21 |
| | Out-of-bounds penalty | -10.0 |
| | Collision penalty | -20.0 |
| | Target reached bonus | 10.0 |
| | Target threshold | 0.1 |
| Initial Sampling | Near target (%) | 10 |
| | Below buffer (%) | 20 |
| | Above buffer (%) | 20 |
| | Random (%) | 50 |
| | Target start radius | 0.5 |
| TD3 Algorithm | Random seed | 40 |
| | Replay buffer size | 2,000,000 |
| | Batch size | 20,000 |
| | Discount ($\gamma$) | 0.99 |
| | Target update rate ($\tau_{\text{TD3}}$) | 0.005 |
| | Policy noise | 0.2 |
| | Noise clip | 0.5 |
| | Policy update freq. | 2 |
| | Actor learning rate | $3 \times 10^{-4}$ |
| | Critic learning rate | $3 \times 10^{-4}$ |
| | Exploration noise | 0.1 |
| | Random exploration steps | 1,000 |
| | Training timesteps | 20,000 |

Table 18: Architecture and final-verification hyperparameters for the reinforcement learning experiment.

| Category | Parameter | Value |
|---|---|---|
| Architecture | Actor width | 64 |
| | Actor extra hidden blocks | 3 after input block (4 total) |
| | Critic width | 64 |
| | Critic extra hidden blocks | 3 after input block (4 total) |
| | Activation function | Leaky ReLU (0.01) |
| mPOLICE | Start timestep | 16,000 |
| | Interval (timesteps) | 1,000 |
| | Late-training sign-projection margin | 0.0 |
| | Final hidden check margin | $10^{-3}$ |
| | Final hidden solve margin | $1.5 \times 10^{-3}$ |
| | Output check margin | $10^{-3}$ |
| | Output solve margin | $2 \times 10^{-3}$ |
| | Margin acceptance tolerance | $10^{-6}$ |
| | One-step distance retention ($\rho$) | 0.05 |
| Final Loop | Max iterations | 50 |
| | Steps per iteration | 1,000 |
| | Patience | 50 |
| | Penalty weight | 1.0 |
| | Reward threshold | 4.5 |
| | Fixed admissible evaluation starts | 100 |

## H   DeepONet Experiment Details

The trunk is a two-hidden-layer ReLU MLP of width 128 and output dimension 32. The branch maps nine forcing coefficients to 32 basis weights using hidden width 128. Base training uses Adam for 600 epochs. The mPOLICE arm then uses 300 guided epochs, ADMM sign enforcement at requested margin $10^{-4}$ followed by a tighter per-neuron QP solve, output projection, and 300 recovery epochs training the branch and trunk output while hidden trunk layers remain frozen. Soft baselines instead receive 300 fine-tuning epochs with $\lambda \in \{10, 100, 1000\}$. Training uses 500 forcing functions with 500 sampled coordinate queries per function. Evaluation uses 50 unseen forcings. For forcing $f$, relative $L^2$ error is $\|u_\theta - u\|_{2,G}/\|u\|_{2,G}$ on the full $64 \times 64$ grid $G$. Over the 600 sampled pad points, the other columns report the maximum and mean of $|u_\theta(f, y)|$. The table averages these per-forcing metrics over one deterministic training seed (42).

Table 19: DeepONet results for the Poisson operator with two internal Dirichlet pads ($u = 0$), from one trained model evaluated on 50 test forcings. Mean and standard deviation are across forcings, not training seeds.

| Method | Rel-$L^2$ (mean $\pm$ std) | Per-forcing max (mean) | Pointwise violation (mean) |
|---|---|---|---|
| Base (no constraint) | **0.0203 ± 0.0098** | $5.75 \times 10^{-3}$ | $5.90 \times 10^{-4}$ |
| Soft Penalty ($\lambda$=10) | 0.0494 ± 0.0172 | $6.97 \times 10^{-4}$ | $7.95 \times 10^{-5}$ |
| Soft Penalty ($\lambda$=100) | 0.0499 ± 0.0173 | $2.71 \times 10^{-4}$ | $3.34 \times 10^{-5}$ |
| Soft Penalty ($\lambda$=1000) | 0.0916 ± 0.0251 | $3.33 \times 10^{-5}$ | $1.37 \times 10^{-5}$ |
| mPOLICE | 0.0880 ± 0.0276 | **$2.43 \times 10^{-17}$** | **$6.49 \times 10^{-18}$** |

## I   Neumann Boundary-Strip Example

As a complement to Robin, a left-boundary segment is thickened to $R^\varepsilon = [0, \varepsilon] \times [y_{\min}, y_{\max}]$. In the single seed-42 run, mPOLICE has flux MSE $2.7 \times 10^{-23}$, maximum dense flux error $1.5 \times 10^{-11}$, and interior MSE $9.4 \times 10^{-5}$. Numerical verification of the returned model checks positive signed margins, finiteness, uniqueness, and a boundary-functional residual below $10^{-9}$ (measured $1.5 \times 10^{-11}$). Under the theorem's exact-arithmetic hypotheses, the same residual bound extends over the covered face. The soft-penalty run

Figure A3: Neumann boundary-strip example. The top row shows the FD reference, soft-penalty solution, mPOLICE solution, and absolute error. The middle row shows the flux along the strip and penalty-weight sensitivity. The bottom row shows error summaries.

with $\lambda = 0.1$ fits the interior better $(3.7 \times 10^{-6})$ but has flux MSE $1.1 \times 10^{-3}$. The best tested soft flux MSE is $6.0 \times 10^{-5}$ at interior MSE $1.5 \times 10^{-4}$.

## J  Diffusion with a Partially Covered Internal Wall

A seed-42 run studies steady diffusion on $[0, 1]^2$ with Dirichlet data on the left and right, homogeneous Neumann data on the top and bottom, and a rectangular insulating obstacle. Four face strips are inset from their shared corners to maintain strict distinct-pattern margins. The resulting construction covers most, but not all, of the closed wall.

Four thin strips cover approximately 96% of the obstacle boundary and are inset by 0.01 at the corners. Natural mean patterns are enforced by ADMM followed by a tighter QP solve at requested margin $10^{-4}$, and the output projection imposes zero face-normal functionals. Numerical verification of the returned model reports minimum signed margin $9.98 \times 10^{-5}$, unique patterns, finite parameters, maximum face-functional residual $7.8 \times 10^{-14}$, and an independent covered-strip autograd check of $1.49 \times 10^{-13}$. Under the theorem's exact-arithmetic hypotheses, the covered-strip functional is constant. The returned values are numerical

Figure A4: Partially covered wall experiment. The top shows the FD reference, an underfit soft-penalty run, and mPOLICE. The bottom shows absolute errors and the centerline slice. Numerical verification of mPOLICE reports a covered-strip flux diagnostic below $1.5 \times 10^{-13}$. Under the theorem's exact-arithmetic hypotheses, that residual extends over each covered strip. The strips cover only $\approx 96\%$ of the wall, and the corner-gap diagnostic reaches 2.52. The unequal-fit soft comparison is qualitative.

residuals rather than exact zeros. The uncovered $\approx 4\%$ has no guarantee and its measured full-face diagnostic reaches 2.52.

Figure A4 also shows one soft-penalty run. Its active-domain MSE is $1.66 \times 10^{-1}$ after 250 fine-tuning epochs, versus $6.39 \times 10^{-4}$ after mPOLICE's 500 full-network and 600 output-only recovery epochs. Because the training budgets and fit differ, the missing temperature shadow can reflect global underfitting as well as the measured covered-strip flux of $5.3 \times 10^{-2}$. This unequal-fit comparison is qualitative.

# K   Implicit Occupancy-Field Learning with Geometric Constraints

In this experiment, a neural network learns an implicit 3D occupancy field for an extruded stadium with two cylindrical holes. The model first becomes affine on each inscribed convex 30-gonal cylinder and then imposes zero occupancy at its vertices. Conditional on the recorded positive-margin affine-cell check, the vertex residual provably bounds the field throughout those polygonal cylinders. The annular slivers between the inscribed polygons and the target circular holes lie outside this guarantee.

The method uses known task geometry. Because the target is extruded, all first-layer $z$-coordinate weights are fixed to zero. Forty fixed first-layer half-space features (20 per hole) and two fixed second-layer polygonal-bump

Table 20: Matched seed-42 3D occupancy comparison. Both rows use the same fixed hole features and extrusion symmetry. "Hole max." is the largest absolute prediction over sampled hole voxels. Topology means one connected body and exactly two enclosed holes on every sampled voxel-depth slice.

| Method | Overall MSE | IoU | Dice | Hole max. | Topology |
|--------|-------------|-----|------|-----------|----------|
| Matched reference (unconstrained) | $3.962 \times 10^{-3}$ | 0.9934 | 0.9967 | 0.6667 | Yes |
| Geometry-guided mPOLICE | $4.436 \times 10^{-3}$ | 0.9782 | 0.9890 | $5.83 \times 10^{-4}$ | Yes |

features localize the holes. The other rows of the width-320 network are trained on a filled-stadium auxiliary target, their mean-assigned patterns are projected once by a double-precision per-neuron QP at solve margin $1.2 \times 10^{-4}$, and a rank-reduced equality-constrained ridge readout ($\lambda = 10^{-8}$) is fit to the original labels. The matched unconstrained reference uses the same fixed features and extrusion symmetry but trains on the original holed target. Table 20 compares this complete constrained pipeline with its matched unconstrained reference. The comparison evaluates the pipelines as configured rather than isolating the readout.

After serialization and reload, numerical verification reports minimum signed margin $1.1987 \times 10^{-4}$ (acceptance margin $10^{-4}$), unique region patterns, finite parameters, and maximum unscaled vertex residual $1.10 \times 10^{-12}$. Under the theorem's exact-arithmetic hypotheses, the same residual bound extends throughout each inscribed polygonal cylinder. The equality system has rank 6 and 315 readout null-space degrees of freedom. On the sampled voxel grid, the checkpoint preserves the desired topology on every depth slice. Its maximum hole-area relative error is 0.122, computed across slices after pairing enclosed holes by descending voxel area, and its overall MSE is modestly worse than the matched unconstrained reference.

Figure A5 shows the target, the matched unconstrained reference, and the numerically verified checkpoint. Magenta marks false-positive solid voxels at the 0.5 occupancy threshold.

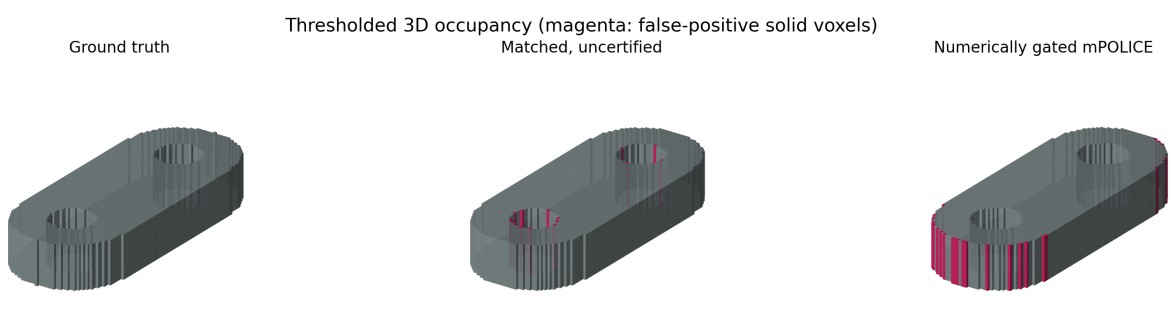

Thresholded 3D occupancy (magenta: false-positive solid voxels)
Ground truth      Matched, uncertified      Numerically gated mPOLICE

Figure A5: 3D occupancy comparison at threshold 0.5 for the target, matched unconstrained reference, and numerically verified mPOLICE. Magenta denotes false-positive solid voxels. On every sampled voxel-depth slice, the verified checkpoint preserves one connected body and both holes (MSE $4.44 \times 10^{-3}$, IoU 0.978, Dice 0.989). The provable regional guarantee covers the inscribed polygonal cylinders, not the full circular-hole boundaries.

## L  Robin Boundary Condition Experiment Details

The domain is $[0,1]^2$ discretized on a $101 \times 101$ finite-difference grid. The temperature field satisfies $-\nabla^2 T = Q(x,y)$ with a localized Gaussian heat source centered at $(0.5, 0.5)$. Boundary conditions are $T=0$ on $x=1$, $y=0$, $y=1$, and the portion of $x=0$ outside $[0.3, 0.7]$. On the cooling-wall segment ($x=0, y \in [0.3, 0.7]$), a Robin condition $hT + k\,\partial_n T = hT_{\mathrm{amb}}$ applies with convective coefficient $h=5$, conductivity $k=1$, and ambient temperature $T_{\mathrm{amb}}=0$.

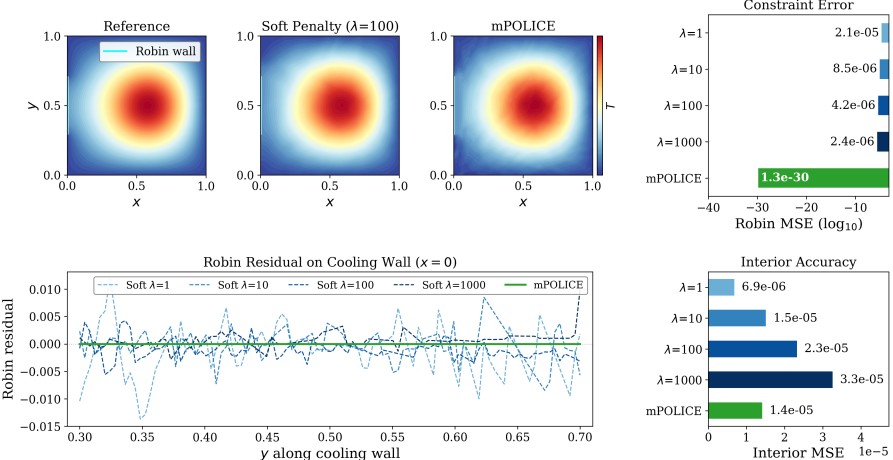

Figure A6: Robin example (convective heat transfer). The top row shows the reference, soft-penalty ($\lambda$=100), and mPOLICE solutions with constraint error. The bottom row shows the Robin residual along the cooling wall and interior MSE.

The network is a 3-hidden-layer ReLU MLP with width 256. Base training runs for 800 epochs (learning rate $10^{-3}$, batch size 256) on supervised samples from the finite-difference reference. For soft-penalty baselines, a Robin-residual penalty $\lambda\|h\,T + k\,\partial_n T - h\,T_{\mathrm{amb}}\|^2$ is added during a 400-epoch fine-tuning phase (learning rate $10^{-4}$), sweeping $\lambda \in \{1, 10, 100, 1000\}$. For mPOLICE, the wall segment is thickened into a strip $R^\varepsilon = [0, 0.05] \times [0.3, 0.7]$. Sign-pattern enforcement uses ADMM at margin $\delta = 10^{-4}$ with a tighter per-neuron QP refinement, and the Robin linear relation is imposed via output-layer projection. The returned network undergoes the shared numerical verification plus the Robin vertex-functional residual check. Unlike the soft baselines, mPOLICE's recovery phase adds no Robin penalty. After enforcement and projection, all hidden layers are frozen (preserving the checked sign pattern) and the output layer alone is fine-tuned on the task loss for 1,500 epochs at learning rate $10^{-3}$, re-projecting the Robin relation every 25 epochs and once more before final verification.

## M   Synthetic Zero-Region Field Regression

This synthetic coordinate-regression target is derived from a scalar XLB (Ataei & Salehipour, 2024) velocity-magnitude array. Two $16 \times 16$ squares are set to zero before training. The task fits this modified field with zero-output constraints on the squares. It is not a simulation of new obstacles or no-slip flow.

Table 21: Seed-31 synthetic field-regression run. Times cover method-specific phases on the recorded CUDA device. They characterize these runs rather than controlled throughput. Numerical verification of mPOLICE takes a further $0.13\,\mathrm{s}$.

| Method | Overall MSE | Base (s) | Additional phase (s) | Numerical verification |
|---|---|---|---|---|
| Soft penalty, from scratch | $2.0 \times 10^{-5}$ | — | 114.78 | Not run |
| mPOLICE, ADMM + QP | $1.1 \times 10^{-5}$ | 97.56 | 159.29 | Pass |

The mPOLICE arm trains a width-128, three-hidden-layer ReLU coordinate MLP for 500 base epochs, performs 200 epochs of projected training with output projection, then 200 epochs of output-layer recovery. Max-margin ECOC assignment is followed by ADMM and a double-precision QP refinement at requested margin $10^{-4}$. Numerical verification of the returned in-memory model reports minimum signed margin $1.285 \times 10^{-4}$, unique patterns, finite parameters, and maximum equality residual $2.745 \times 10^{-13}$ in standardized output ($4.53 \times 10^{-15}$ in original scalar units). Under the theorem's exact-arithmetic hypotheses, the same vertex-residual bound extends throughout both squares. The soft-penalty run uses a separate initialization

and 500 epochs with vertex penalty weight 1000, without positive-margin verification. Its values report task fit only, while the regional verification applies to mPOLICE.

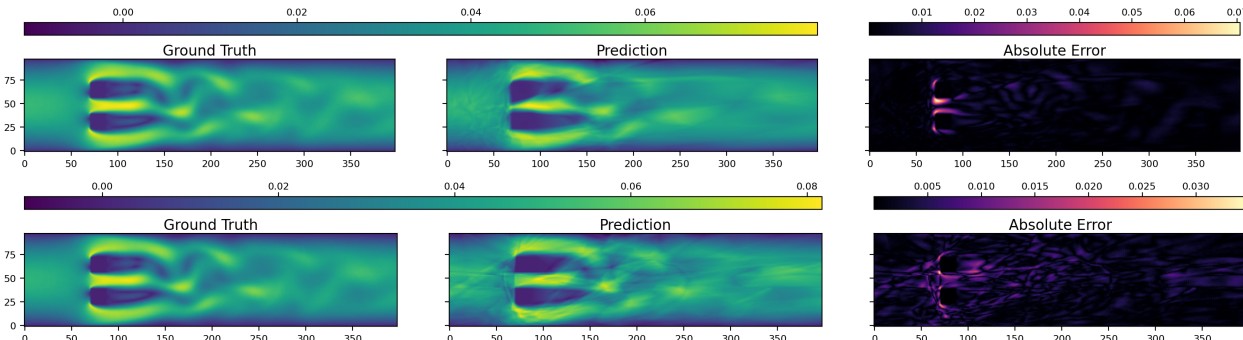

Figure A7: Seed-31 synthetic zero-region field regression corresponding to Table 21. Each strip shows the modified XLB-derived target, the prediction, and the absolute error. The upper strip is the soft-penalty run and the lower strip is mPOLICE. Color scales are shown separately for each strip. The images illustrate task fit, while the region-wide numerical verification reported in the text applies only to mPOLICE.

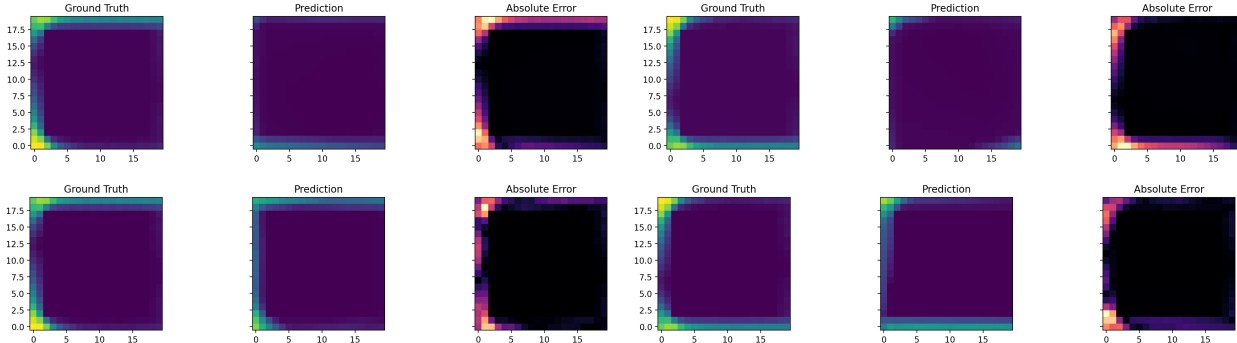

Figure A8: Close-ups of the two constrained squares for the same seed-31 run. The columns correspond to the two squares. The upper row shows the soft-penalty run and the lower row shows mPOLICE. Each triplet contains the modified target, the prediction, and the absolute error. Target and prediction share a scale within each triplet, while the error is normalized separately.

# N   Performance Benchmark Details

The performance benchmark in Section 4.5 uses a frozen four-hidden-layer, width-64 ReLU MLP, 32-vertex regular polygons, region counts $\{1, 2, 4\}$, and three seeds. Table 22 reports the four-region setup.

Table 22: Time to a numerically verified result for a frozen $4 \times 64$ ReLU MLP with 4 regular-polygon regions and 32 vertices per region (128 total). All backends use the identical feasible one-vs-rest sign map, requested hidden margin $\delta = 10^{-3}$, output tightening $\eta = 10^{-4}$, and output tolerance $10^{-9}$. Total includes assignment, affinity enforcement, output projection, and final numerical verification. A mean is shown only when all 3/3 seeded runs pass. Failures are shown explicitly and not averaged. The last column times one 4096-query forward batch. The CPU is an AMD EPYC 9B45. Software comprises PyTorch 2.9.0+cu128, CVXPY 1.6.5/CLARABEL 0.11.1 for the common output QP, and OSQP 1.1.3 for hidden-layer QP, using float64 and 1 thread.

| Backend | Assign (s) | Affinity (s) | Output (s) | Verify (s) | Total (s) | 4096-query batch (ms) |
|---------|-----------|-------------|-----------|-----------|----------|----------------------|
| POCS | $0.0031 \pm 0.0000$ | $0.012 \pm 0.000$ | $0.008 \pm 0.001$ | $0.001 \pm 0.000$ | $0.025 \pm 0.001$ | $3.32 \pm 0.11$ |
| ADMM | $0.0031 \pm 0.0000$ | $0.793 \pm 0.486$ | $0.008 \pm 0.001$ | $0.001 \pm 0.000$ | $0.806 \pm 0.487$ | $3.32 \pm 0.12$ |
| QP | $0.0031 \pm 0.0000$ | $0.881 \pm 0.193$ | $0.008 \pm 0.001$ | $0.001 \pm 0.000$ | $0.894 \pm 0.193$ | $3.32 \pm 0.12$ |

The network receives eight warm-up epochs on 1500 regression samples and is then frozen. Region centers are vertices of a regular polygon, making the prescribed first-layer one-vs-rest identifiers linearly separable. Every backend receives that identical sign map, hidden margin $\delta = 10^{-3}$, output tightening $\eta = 10^{-4}$, and strict CLARABEL output projection. The benchmark measures enforcement overhead, not end-to-end training.

For each seed and region count, we time one complete mPOLICE call,

$$T_{\text{total}} = T_{\text{assign}} + T_{\text{affinity}} + T_{\text{output}} + T_{\text{verify}},$$

where the terms time sign-map construction, hidden-layer enforcement, output projection, and shared final verification. A row is successful only if the returned model passes full numerical verification, attains the requested margin within $5 \times 10^{-7}$, and satisfies the tightened output check. Failures are reported explicitly and excluded from partial averages. All 27 runs pass. One four-region QP seed requires two bounded layerwise passes, both included in its time. We also time a batch of 4096 forward queries. The inference graph is unchanged.

Solver iteration counts make ADMM/QP timings non-monotone across this small region sweep, so the plot is descriptive rather than a fitted complexity law. Section P.1 gives the asymptotic accounting, while E8 and E9 isolate vertex scaling under different protocols.

## O  Enforcement Backend Formulations, Geometry, and Selection

This appendix details the three projection backends summarized in Section 2.6.

**Quadratic programming (QP).**  Fix a layer $\ell$ and a neuron $n$ with current parameters $(\mathbf{w}_n^{(\ell)}, b_n^{(\ell)})$. For each region $R_i$ with vertices $\{\boldsymbol{v}_p^{(i,\ell)}\}$ and prescribed sign $\text{sign}_n^{(i,\ell)} \in \{\pm 1\}$, enforcing a margin $\delta \geq 0$ amounts to the half-space conditions

$$\text{sign}_n^{(i,\ell)} \left( \mathbf{w}_n^{(\ell)\top} \boldsymbol{v}_p^{(i,\ell)} + b_n^{(\ell)} \right) \geq \delta, \quad \forall p, i.$$

Seeking the smallest change from the current parameters in Euclidean norm yields, for each neuron,

$$\min_{\Delta \mathbf{w}_n^{(\ell)}, \Delta b_n^{(\ell)}} \quad \left\| \Delta \mathbf{w}_n^{(\ell)} \right\|_2^2 + \left( \Delta b_n^{(\ell)} \right)^2$$

$$\text{s.t.} \quad \text{sign}_n^{(i,\ell)} \left( (\mathbf{w}_n^{(\ell)} + \Delta \mathbf{w}_n^{(\ell)})^\top \boldsymbol{v}_p^{(i,\ell)} + (b_n^{(\ell)} + \Delta b_n^{(\ell)}) \right) \geq \delta, \ \forall p, i.$$

If the half-space intersection is nonempty, this strictly convex QP has a unique exact-arithmetic solution. It is the minimal Euclidean perturbation attaining the requested margin. The numerical implementation returns an approximation to that solution at the solver tolerance, so the attained margin is re-evaluated rather than inferred from solver status.

**Projection onto Convex Sets (POCS).** Because each sign constraint is a half-space in the augmented parameter vector $\theta = [\,\mathbf{w}^\top \; b\,]^\top$, we can also use standard POCS. It repeatedly projects onto the violated half-spaces until the iterates stabilize. The update is simple, numerically stable, and often converges in a few sweeps when only a small subset of vertex constraints is active. We use it as a lightweight projection backend.

**Batched ADMM projection.** We also employ standard ADMM splitting (Gabay & Mercier, 1976) to solve the Euclidean projection layerwise. By stacking vertex constraints, ADMM alternates between a quadratic parameter update and projection onto the shifted orthant $z \geq \delta\mathbf{1}$. Since the linear system matrix is shared across neurons in a layer, its factorization can be reused, offering a scalable solver for this subproblem.

**Geometry of the backends and how to choose one.** All three backends target the *same* intersection of margin-$\delta$ half-spaces. The shared guarantee comes from re-evaluating the returned point against the margin criterion, not merely from convergence status or backend choice. The QP computes the Euclidean projection to solver tolerance for each neuron's half-space subproblem, with layers processed sequentially. This gives a per-neuron minimal parameter change, not a globally minimal-norm network repair. ADMM targets that projection asymptotically but can return a feasible point farther from the initial parameters at practical iteration budgets. In E9, this distance is approximately $1.9\times$ the QP distance.

Standard cyclic POCS converges to a feasible point when the intersection is nonempty, but that point is not generally the nearest one. Our implementation selects the most violated half-space and accepts its result only after the common margin check. In E9, its movement norm matches QP to two decimal places. Downstream differences therefore stem from stopping tolerances and the selected feasible point, not different constraint sets.

E9 compares the backends and a POCS→QP hybrid under identical margins and sign assignments. All returned points pass the same signed-margin check. POCS and ADMM run in milliseconds per cycle up to thousands of constrained vertices. The hybrid runs inexpensive POCS sweeps and then a QP from the POCS iterate, so its minimal-change property is relative to that iterate rather than the original parameters. Feasibility is established by the post-solve check. We use ADMM when factorization reuse or $\delta = 0$ matters, POCS when per-cycle speed matters, and QP when the per-neuron minimal-change property matters.

# P  Complexity, Termination, and Verification Details

## P.1  Time Complexity and Termination

Let $W$ be the common hidden width, $L$ the total number of affine layers, $D$ the input dimension, $K$ the output dimension, $H = (L-1)W$ the hidden-neuron count, $P = \sum_i P_i$ the constrained-vertex count, and $b = \lceil \log_2 N \rceil + r$ the identifier-bit count. mPOLICE prescribes cells rather than enumerating them. Sign assignment costs one hidden-vertex forward pass, $O(PDW + P(L-2)W^2)$, plus greedy ECOC matching $O(N^2 b)$ in the usual path. Checking complete patterns costs $O(N^2 H)$. Conditional on fixed propagated vertices and signs, enforcement solves $W$ convex projections per layer. For a layer with input width $n$ and $J$ fixed backend iterations, ADMM costs $O(Pn^2 + n^3 + JWn(P+n))$ and POCS costs $O(JPWn)$. QP runs $W$ solver calls. Each reported solver call has a finite iteration or time cap. These caps ensure termination of a call but do not bound its iteration complexity or guarantee success. The output problem has $K(W+1)$ variables, with cost also depending on its constraint-row count. These expressions describe per-call costs in the explicit dimensions and do not establish polynomial-time complexity for the joint nonconvex procedure. Moreover, $P$ can itself be exponential in $D$ for a V-represented box. Solver, iteration, epoch, and retry caps guarantee procedural termination, not success, so numerical claims are reported only when checks of measured margins, uniqueness, residuals, and finiteness pass. Additional costs and the verification scope are detailed below.

**The joint sign-assignment problem.** The following display is a *schematic summary* of the region-consistency and uniqueness requirements of Section 2.4:

$$\min_{\boldsymbol{\theta},\,\{\mathrm{sign}_n^{(i,\ell)}\}} \quad \Phi(\boldsymbol{\theta}) \quad \text{subject to}$$

$$\mathrm{sign}_n^{(i,\ell)}\left(\mathbf{w}_n^{(\ell)\top}\boldsymbol{v}_p^{(i,\ell)} + b_n^{(\ell)}\right) \;\geq\; \delta, \;\; \forall p, i, n, \ell,$$

$$\forall\, i \neq j,\; \exists\, n, \ell \text{ such that } \mathrm{sign}_n^{(i,\ell)} \;\neq\; \mathrm{sign}_n^{(j,\ell)},$$

Here $\Phi(\boldsymbol{\theta})$ is the task objective and every sign variable lies in $\{\pm 1\}$. The first line represents region consistency and the second pairwise pattern uniqueness. Distinct realized cells additionally require a positive attained margin. This display is not a complete mixed-integer convex program. The vertices $\boldsymbol{v}_p^{(i,\ell)}$ depend recursively on the parameters in layers $1, \dots, \ell-1$, and the existential uniqueness condition would need auxiliary binary variables and a chosen encoding. The resulting joint task is combinatorial and non-convex. In practice, signs are chosen heuristically and, conditional on the propagated vertices and fixed signs at one layer, the corresponding half-space projection is convex.

**Layer totals and first-layer terms.** For a hidden layer with input width $n$, ADMM constraint-matrix assembly is $O(Pn^2)$ and one $O(n^3)$ Cholesky factorization is shared across its neurons and iterations. With $J$ solver iterations, the residual products and solves cost $O(JWn(P+n))$. Across the $L-2$ hidden layers whose input width is $W$, this gives $O((L-2)(W^3 + JW^2(P+W)))$, plus the first-layer cost $O(PD^2 + D^3 + JWD(P+D))$. POCS costs $O(JPWn)$ per layer. The per-neuron QP solves $W$ convex QPs per layer in $n+1$ variables each using OSQP in our implementation. Finite solver caps ensure termination of each call but not success within the cap. The input dimension $D$ also enters through the user-chosen vertex count $P$, which can be exponential in $D$ for full-dimensional V-represented regions.

**Max-margin ECOC scoring.** With $m_{n,i}^+ = \min_p z_n(\boldsymbol{v}_p^{(i,\ell)})$ and $m_{n,i}^- = -\max_p z_n(\boldsymbol{v}_p^{(i,\ell)})$, every neuron is scored by $\min_i \max(m_{n,i}^+, m_{n,i}^-)$, and the $b = \lceil \log_2 N \rceil + r$ highest-scoring neurons become identifier bits. Codewords are generated subject to distinctness and a requested Hamming separation when possible, then greedily matched to regions by nearest natural signs. This is not a globally optimal assignment. Remaining neurons take their larger-margin sign.

**Implementation overheads.** Several costs sit on top of the solves. Re-propagating vertices from the input for every target layer (and again for margins) adds $O(PLDW + PL^2W^2)$. Caching would remove the extra $L$. Greedy region–codeword matching costs $O(N^2 b)$ bit comparisons, and the complete-pattern uniqueness check costs $O(N^2 H)$. Bounded random codeword generation adds $O(ANb)$ for $A$ attempts, and, for $b \leq 16$, failure to find enough separated words can trigger a deterministic scan costing up to $O(2^b Nb)$. After the vertex outputs are evaluated, a sparse stacked residual check is linear in the stored nonzeros of $\mathcal{A}$ and $\mathcal{C}$. These are implementation costs, not a guarantee that the chosen discrete assignment is feasible.

**Requested vs. attained margins in sweep experiments.** The shared verification routine uses the strict margin check $m > 0$ from Theorem 1. Attaining the *requested* margin, $m \geq \delta - \epsilon_{\mathrm{solve}}$, is Algorithm 1's separate acceptance rule. Sweep experiments that report single bounded solves record the attained margin as a measurement instead of enforcing that rule. E9's vertex-timing sweep and E10's capacity sweep mark such shortfalls explicitly.

**Per-experiment numerical verification.** The shared routine applies to networks with linear output activations and evaluates signed margins, pairwise pattern uniqueness, equality/inequality residuals, and finiteness on returned parameters. Full verification is applied to E4, E5, E9–E11 (E11 waives uniqueness by design), E13, the paper-scale DeepONet, Robin, Neumann, wall, performance, 3D-shape, and synthetic-field runs. Robin/Neumann add functional residuals, the wall adds an independent dense-flux check, and performance also requires the requested margin and tightened output residual. RL combines the full sign checks with protected-set and predecessor-guard geometry, task-specific slab inequalities, clamp preservation, transition conditions, a dense one-step diagnostic, and checkpoint-reload verification. E1/E7 use task-specific

checks, E6 residual-only, E12 sign-only, E8 full verification outside each timer, and E2 full verification after every re-projection. The APRNN adapter analogously checks each stage. Table 1 and E3 report diagnostics without full numerical verification.

**Floating-point scope.**   The tolerance $\varepsilon$ in Figure 2 is a stopping heuristic only. The reported verification values come from ordinary floating-point evaluations, not conservative interval bounds. Most runs use double precision. The returned RL checkpoint and its task-specific checks use float32 and are repeated after reload. Passing configurations choose $\delta$ and $\eta$ well above machine epsilon, while $\delta = 0$ configurations have no strict positive-margin verification (Tables 10–11). The checks report computed margins and residuals for the returned floating-point model but do not prove exact equalities or inequalities under floating-point execution. A formal finite-precision proof would require directed-rounding interval arithmetic or rational verification.

