# OpenReview forum: "Provable Multi-Region Affinity Enforcement and Constraint Satisfaction for Scientific Machine Learning"
_TMLR — Under review for TMLR_

### Review · Reviewer_AzNy · 2026-05-25

**Summary Of Contributions:**

This paper studies the problem of enforcing hard constraints for scientific ML. In particular, given a piecewise-liner DNN, multiple pairs of input convex regions and output constraints, this paper presents mPOLICE, a framework to modify the DNN parameters so that the modified DNN is guaranteed to satisfy the given output constraints for all inputs in the corresponding input convex regions.

mPOLICE achieves this via a two-stage algorithm. In the first stage, mPOLICE modifies the piecewise-linear DNN so that the DNN is locally affine to each input convex region. Then in the second stage, mPOLICE incorporates the output constraints to minimize violation error while maintaining the affinity enforcement.

This paper experimentally evaluated mPOLICE on a variety of scientific ML tasks and properties, operator learning, PDE boundary-condition enforcement, reinforcement-learning safety constraints, etc., demonstrated the effectiveness of mPOLICE compared to soft-penalty baselines, and practicality in relative small-scale scientific ML DNNs.


## Strength

- This paper studies an important problem: enforcing hard constraints for scientific ML, where soft penalty methods fail to deliver as evidenced by experiments and existing work.
- This paper presents a practical implementation to the affinity enforcement and constraint enforcement stages.
- The proposed method provides provable guarantee, no inference-time overhead and no architecture changes, and can be integrated with standard training pipelines.
- The experiments cover a variety of scientific ML tasks and properties, and demonstrate the effectiveness of mPOLICE compared to soft-penalty baselines, and practicality in relative small-scale scientific ML DNNs.

## Weaknesses

- The current paper's contribution and positioning claims are not supported by an accurate and clear account of existing hard-constraint enforcement work, especially the closely related PLDI 2023 paper (see below).
- The use of violation thresholds may break the soundness of the provable guarantee.
- Lack of discussion on the convergence and complexity in theory and practice of the proposed algorithms, in terms of the size of input regions and the DNN.
- Limited applicability: mPOLICE is restricted to first-order equations, linear boundary conditions and piecewise-linear DNNs, which limits its use in most scientific ML tasks.

**Additional Comments:**

None.

**Audience:**

Yes

**Audience Explanation:**

TMLR's audience interested in scientific machine learning, formal methods for AI and trustworthy AI would be interested in the findings of this paper. Enforcing hard constraints is an important problem for scientific ML, and this paper presents a practical method with a variety of scientific ML applications.

However, the paper's contribution and positioning claims are not supported by an accurate and clear account of closely-related existing work. A clear review would help the TMLR audience understand the actual contribution of this paper.

**Broader Impact Concerns:**

None.

**Claims And Evidence:**

No

**Claims Explanation:**

## Issue 1: The current paper's contribution and positioning claims are not supported by an accurate and clear account of closely-related existing work.

The abstract states that "while existing hard-constraint methods are typically specialized to particular equations or geometries", and the introduction mainly reviews such specialized hard-constraint methods. This background narrative is incomplete and potentially misleading because it omits a closely related, more general prior work: Tao et al., "Architecture-Preserving Provable Repair of Deep Neural Networks," PLDI 2023.

Later in the related work section, the PLDI 2023 paper is characterized as a different "target output containment" line of work, and does not control neuron activation patterns. In my view, this characterization is inaccurate: the PLDI 2023 paper is closely related to the central contribution of this paper, and its core is controlling neural activation patterns. In particular, the PLDI 2023 paper:

1. **Addresses the same problem** of enforcing output constraints for multiple disjoint vertex-represented input V-polytopes (regions in this paper) on DNNs by only modifying the DNN parameters while preserving the DNN architecture.
2. **Proposes the same key insight** called local linearity (local affinity in this paper) to reduce the constraints on the input convex regions to constraints on the vertices by controlling the neural activation patterns.
3. **Proposes a similar two-stage algorithm**: first control the neural activation patterns to "shift" its linear pieces so that the DNN becomes locally linear to each input V-polytope, and then "assert" the output constraints while maintaining the local linearity. This corresponds to the two stages of this paper: first enforce local affinity, then enforce constraints.
4. Also claims provable guarantee, no inference-time overhead and no architecture changes.

For better understand the connection between the two papers, here are some concepts referred differently in the PLDI 2023 paper vs this paper:

- "Provable V-polytope repair" vs "Provable Multi-Region Affinity Enforcement and Constraint Satisfaction"
- "V-polytope" vs "region"
- "Locally linear" vs "locally affine"
- "Linear piece/region" vs "activation cells"
- Stage 1: "shift" vs "affinity enforcement"
- Stage 2: "assert" vs "constraint enforcement."

That said, there are meaningful differences that could form the basis for a clearer contribution claim. For example:

- While the PLDI 2023 reduces both stages to linear programming problems for scalability, this paper implements the two stages using QP and training-based methods.
- This paper focuses on scientific ML applications, while the PLDI 2023 paper focuses on general DNN applications.

## Issue 2: Unsound use of violation thresholds breaks the provable guarantee of the method.

The use of violation thresholds in this paper raises my concern in soundness. For a hard-constraint enforcement method to be provable, numerical tolerance must be handled very carefully. An absolute or relative violation may be acceptable for equality constraints in regression tasks, but is not generally acceptable for hard qualitative properties like positivity, boundedness, etc. For such properties, a violation within a small threshold on the wrong side is still a violation.

This issue is especially serious for the local affinity enforcement constraints. The soundness of the proposed method relies on the modified DNN being locally affine to each input convex region. However, if the sign assignment is violated even within a small threshold, new vertices could be created and the convex hull of the outputs on the input vertices may no longer contain outputs for all inputs in the convex region, which breaks the provable guarantee of the method.

**Requested Changes:**

1. The paper should be revised to include a clear and accurate review of existing work, especially the PLDI 2023 paper, and clearly position the contribution of this paper in relation to existing work.
2. The paper should clarify the use of violation thresholds, clearly state the conditions under which the formal guarantee remains valid, and provide a sound way to handle numerical tolerance to maintain the provable guarantee of the method.
3. The paper should include a discussion on the convergence and complexity in theory and practice of the proposed algorithms, in terms of the size of regions and the DNN.

---

> ### Author Response · Authors · 2026-07-15
> **Response to Reviewer AzNy**
>
> Thank you for this review. We address the three requested changes in order, citing the new controlled Experiments E0–E13 (Appendices A–B, reproducible code included in the SM).
>
> ## 1. Positioning relative to APRNN (Issue 1)
>
> We agree that better positioning was needed. The revised Related Work (Section 1.1) now presents APRNN as a close prior work and describes it accurately. It preserves the architecture, controls activation patterns so each target V-polytope lies in a locally affine piece, and reduces whole-polytope conditions to vertex relations via its Shift/Assert stages. The earlier "target output containment" characterization is gone, and we do not claim the vertex reduction as new. The stated contribution is now what APRNN does not cover, namely independent multi-region assignment with repair and verification, one formulation spanning value, interface, inequality, and first-derivative constraints, and repeated projection during continued scientific-ML training.
>
> We also added direct evidence in Appendix A. Because the released `sytorch` pipeline needs a Gurobi license that limited the scale we could run, we built an adapter that re-implements VPolytopeRepair, APRNN's released repair routine, following Shift/Assert and the released evaluation objective on CLARABEL/OSQP. E0 validates it stage by stage against released `sytorch` with Gurobi on small cases. Adapter results are always labeled, and the adapter and comparison scripts ship with the supplementary code.
>
> It supports three limited conclusions. First, both workflows reach solver-scale residuals. Second, their different anchors and training protocols preclude a method-level accuracy ranking, and E7 records failures for both. Third, E8 measures a **43×–280× mPOLICE speedup per re-projection cycle** on a four-hidden-layer, width-32 MLP as one region grows from 4 to 256 vertices (0.0040–0.0162 s versus 0.173–4.539 s). Structurally, APRNN's joint LP/QPs grow with vertex count while mPOLICE reuses one layer factorization, which is what repeated projection during training needs. Appendix A gives protocols, results, and architecture-scale diagnostics, with E1–E2 on reduced architectures for control.
>
> ## 2. Conditions for the guarantee and sound tolerance handling (Issue 2)
>
> You are right on both counts. A small wrong-side value is still a violation, and a wrong-side vertex sign can split a region across cells and void the guarantee. The revision addresses this in three parts.
>
> First, when the formal guarantee is valid. Theorems 1–2 (proofs in Appendix C) are exact-arithmetic statements with no acceptance threshold. The guarantee holds exactly when their hypotheses do, namely common weak vertex signs across each region, strictly positive margins on the distinguishing neurons, and residuals affine in the local map.
>
> Second, sound tolerance handling. Algorithm 1 separates `tau` for assignment, `delta` for the requested sign margin, `eta` for inequality tightening, and `epsilon` for optional stopping, and no check accepts a wrong-side value as small. Verification requires the weak signs to hold as signs and the distinguishing margins to satisfy the strict condition $m > 0$. Inequalities are tightened by `eta` before enforcement, so acceptance means the original property holds with measured slack, never a tolerated violation. Configurations claiming numerical verification return FAIL when their checks fail after bounded repair or polishing. Diagnostic or gate-off runs are labeled and claim nothing stronger.
>
> Third, the effect in practice. In E4, solver-scale misses of the tightened targets still leave about $10^{-3}$ measured vertex slack for the original positivity and boundedness properties, which extend region-wide under the theorems' hypotheses. The float32 RL actor is re-checked after serialization and reload. We reran the experiments under these gates to ensure the reported constraints are provably satisfied.
>
> ## 3. Convergence and complexity in region and network size (Issue 3)
>
> As requested, Appendix P now accounts for assignment, hidden enforcement, output projection, and verification in the number of regions, the user-supplied vertex count, and the network depth and width. Each stage's per-call cost is polynomial in these counts under capped solver iterations. We claim no bound for the joint procedure, whose sign assignment is combinatorial and whose bounded solves can fail. Solver, epoch, iteration, and retry caps guarantee termination, not convergence, and guarantee-bearing results must pass their final verification gates. In practice, Section 4.5 reports measured scaling, with E5 and E8 sweeping regions and vertices, E6 sweeping network size, and E9 comparing backends.
>
> On applicability, Section 5 now states the scope plainly. The framework covers fully connected ReLU or two-slope Leaky ReLU MLPs with affine outputs, disjoint convex polytopes, and residuals affine in the local map. Nonlinear and second-order conditions need new machinery.

---

### Review · Reviewer_E3gL · 2026-05-28

**Summary Of Contributions:**

This submission extends prior work [1] to develop an algorithm for exact constraint satisfaction with ReLU networks and applies it to several domains, for instance neural operator learning, convective heat-transfer with Robin boundary conditions and reinforcement learning with safety constraints. It finds that compared to soft, regularization-based penalties, it can reduce the violations of constraints, albeit at the expense of modeling performance.

### Strengths

- The general problem of exact (rather than soft) constraint satisfaction in deep learning is interesting and important
- The method is validated on many different problems and domains

### Weaknesses

- While prior work [1] explicitly studied problems where the function class was assumed to be affine, here more general constraint satisfaction problems are considered but a justification or discussion of why enforcing local affinity is valid in this setting is missing.
- The time complexity of the algorithm is not discussed and it is not clear to me how it deals with the exponential number of partition regions of the input space. Maybe for this reason the empirical problems studied have a small input dimensionality (d=2), see questions.
- The structure and presentation of the submission needs to be improved, e.g., to more clearly communicate the different algorithm variants being used

[1] Balestriero, R., & LeCun, Y. (2022). POLICE: Provably optimal linear constraint enforcement for deep neural networks.

**Audience:**

Yes

**Audience Explanation:**

The general problem of exact (rather than soft) constraint satisfaction in deep learning is interesting and important.

**Broader Impact Concerns:**

I have no specific concerns for the broader impact of this paper.

**Claims And Evidence:**

No

**Claims Explanation:**

I am primarily concerned by the claim that the algorithm works for general constraint satisfaction problems and that the algorithm is practical in terms of time complexity beyond small input dimensionality, please see the questions in the "Requested Changes" section.

**Requested Changes:**

> The sign-enforcement projection can be computationally intensive for larger networks or many vertices

- Please report the time complexity of this algorithm as a function of the input dimension $d$, the number of layers $L$, and their width $W$. In particular, a ReLU network partitions the space into exponentially many polytopes which would seem to imply that this method is only feasible for small d/L/W?

> First, how do we guarantee that $f_\theta$ is affine on all of $R_i$ rather than merely at sampled points?

> This is the fundamental trade-off between provable region-wise affinity and unconstrained regression accuracy, and it is inherent to any method that truly makes the function affine rather than merely feasible at evaluation points.

- I am a bit confused by the role of the affinity assumption on each constraint region. These two statements from different parts of the text seem to imply that constraint regions _must_ be affine but I do not see why that is true? At least in my understanding from equation (2) and (3) it does not necessarily follow and seems to be an additional assumption. Could you please clarify to what extent local affinity is a justifiable assumption for general constraint satisfaction problems?


> The complete source code for all experiments is included in the supplementary material; we refer the reader there for full implementation details, hyperparameters, and reproducibility instructions.

- While I very much appreciate that the code is made available, it is not clear to me what the algorithm looks like that you use in your experiments. Figure 2 helps but this needs to be described in the main text and the different variants need to be clearly distinguished. At times it felt like each experiment used a slightly different variation of the algorithm. The paper should be self-contained and not require the reader to study the code to figure out this core part.

- Why do you use POCS in 4.1 and ADMM in 4.2? When should which algorithm be used for sign-pattern enforcement? I am missing a systematic comparison of the different variations you introduce.


> After projection, we optionally fine-tune the output layer

- What are "pretraining" and "finetuning" in the context of this algorithm referring to?

- Why do you not compare to POLICE [1] in your experiments?

> Importantly, [...] the gap narrows with network capacity as finer ReLU partitions more tightly enclose the target regions.

- Do you have empirical evidence for this claim, i.e. is there an experiment where you systematically varied the partitions and saw improved modeling performance?

- Figure 2: How is $\varepsilon$ chosen when running the algorithm? Is it guaranteed to terminate in finite time for a given $\varepsilon$?


- There are many undefined symbols in the equations throughout the text, e.g. C, E, d, f are not defined in equation 2-3, the gammas in equation 4 are undefined.

- What are the green dashed squares in Figure 3? The caption could be a bit more comprehensive here.

### Minor

- It is difficult to assess the implications of the trade-off between general model fit and violations within a task. How much either matters might just depend on a downstream task or how accurate qualitative properties are described. For that reason, illustrating qualitative properties as done in the appendix, e.g. Figure 8 or 9, in my opinion deserves more attention in the main text.

- For consistency, I think the y-axis of the bottom left panel of Figure 3 should be on a log-scale as well.

- The caption of Table 1 states that bold marks the best mean but no method is bold

---

> ### Author Response · Authors · 2026-07-15
> **Response to Reviewer E3gL**
>
> Thank you for the detailed review. We answer each question below, citing the new Experiments E0–E13 (Appendices A–B).
>
> **1. Time complexity in $d$, $L$, $W$, and the exponential cell count.** mPOLICE prescribes one activation pattern per constrained region and never enumerates cells, so the exponential cell count never enters the cost. Appendix P reports explicit per-call costs. With $P$ total constrained vertices, sign assignment costs $O(PdW + P(L{-}2)W^2)$ and POCS enforcement costs $O(JPW^2)$ per hidden layer under capped iterations $J$, with the ADMM and QP forms alongside, so every stage is polynomial in $d$, $L$, $W$, and $P$. The one potentially exponential quantity is $P$ itself, since a full-dimensional box needs $2^d$ vertices. We claim no bound for the joint procedure, whose sign assignment is combinatorial. In E13, low-vertex regions cost 6–12 ms per call as $d$ sweeps from 2 to 64, while full-dimensional regions succeed at $d{=}8$ and fail with a detected output-system incompatibility at $d{=}32$. The practical limit is vertex count and joint feasibility, not $d$, $L$, or $W$ alone.
>
> **2. How is $f$ guaranteed affine on all of $R$ rather than at sampled points?** By a verified vertex condition, not by sampling. Theorem 1 shows that if every vertex of $R$ satisfies the assigned weak sign pattern, convexity places all of $R$ inside one closed activation cell, where the network is exactly affine. Verification checks those vertex signs and Theorem 2's strict distinguishing margins, so the conclusion never rests on sampled evaluations.
>
> **3. Is local affinity an added assumption?** You are right that it does not follow from Equations 2 and 3, and it is not an assumption but a property the algorithm constructs. We enforce it because it is what reduces infinitely many pointwise constraints over a region to finitely many linear vertex conditions, which is its justification. The cost is expressivity around the constrained regions, quantified in E10 and stated as a limitation in Section 5.
>
> **4. The algorithm used in the experiments.** Section 2 now describes the complete workflow (assign and repair patterns, enforce hidden signs, project the output layer, continue training, verify or fail), Algorithm 1 gives pseudocode, and Tables 10–11 list every experiment's configuration and verification mode. The paper no longer requires reading the code.
>
> **5. When to use POCS versus ADMM versus QP.** Use POCS when per-cycle speed matters, ADMM when its shared factorization is reused across repeated re-projections, and QP when a minimal per-neuron change or final polishing matters. Section 2.6 and Appendix O give this guidance, and E9 is the matched comparison. With the checkpoint, assignment, margin, and gate fixed, every backend passes the same verification and differs only in runtime and the returned feasible point.
>
> **6. Pretraining and fine-tuning.** Pretraining is task-only training of the unconstrained network before enforcement. Fine-tuning is the constrained continuation after sign assignment, such as output-layer-only training with periodic re-projection. Section 2 now defines both terms.
>
> **7. Why no comparison with POLICE?** Original POLICE handles one region through bias adjustment and cannot assign distinct patterns to multiple regions, so a direct multi-region run is undefined. We instead added E11, a controlled POLICE-style ablation whose arms share the enforcement and projection machinery and differ only in shared versus distinct patterns. The shared arm becomes affine across the hull spanning both regions, reproducing POLICE's coupling, while mPOLICE stays close to the unconstrained reference there.
>
> **8. Evidence that the gap narrows with capacity.** E10 now sweeps width 16–128 at fixed constraints. The observed fit gap shrinks as width grows, while larger requested margins enlarge the containing cell and hurt fit. Both are measured trends, not guarantees, and the text now ties the claim to E10.
>
> **9. How is $\epsilon$ chosen, and does Figure 2 terminate?** $\epsilon$ is an optional task-scale early-stopping heuristic. Termination is guaranteed for any $\epsilon$ since every solve, epoch loop, and retry has a finite cap. Success is not. A run that exhausts the caps without passing verification returns FAIL, and ordinary floating-point checks are not formal finite-precision certificates (Appendix P).
>
> **10. Notation and minor points.** Section 2.2 now defines $C$, $E$, $d$, and $f$, and the gammas are defined at Equation 4. The green dashed squares mark the internal pads where the Dirichlet constraints are enforced, as the caption states. The bottom-left panel of Figure 3 is now on a log scale, and Table 1's bolding renders correctly.

---

### Review · Reviewer_vkqH · 2026-07-01

**Summary Of Contributions:**

This paper studies hard constraint satisfaction for neural networks in scientific machine learning. The main motivation is that soft-penalty methods can reduce constraint violations but do not guarantee exact satisfaction and often require careful tuning. To address this issue, the authors propose mPOLICE, which exploits the piecewise-affine structure of ReLU networks. By assigning different activation patterns to multiple disjoint target regions, the method enforces the network to be affine on each region. Then, constraints over the whole region can be reduced to linear constraints on the region vertices.
Compared with the original POLICE method, mPOLICE handles multiple constrained regions independently and avoids the convex-hull coupling issue. The method is evaluated on synthetic inequality constraints, DeepONet with internal Dirichlet constraints, Robin boundary-condition enforcement, and safety-constrained reinforcement learning. The results show that mPOLICE can achieve nearly exact constraint satisfaction, although this may come with reduced local flexibility and larger prediction error.

**Audience:**

Yes

**Audience Explanation:**

Yes. I think this paper would be of interest to researchers working on real-world problems where practical considerations are imposed in the form of hard constraints .

**Claims And Evidence:**

Yes

**Claims Explanation:**

The main claim is that mPOLICE can enforce exact constraint satisfaction over multiple disjoint regions by making the network affine on each target region. The experiments generally support this claim, as mPOLICE achieves much smaller constraint violations than soft-penalty baselines, often close to numerical precision. The claim that mPOLICE extends POLICE to the multi-region setting is also reasonable, since assigning different activation patterns addresses the convex-hull coupling problem.
The paper also claims zero inference-time overhead for mPOLICE, which appears plausible because the final model remains a standard ReLU network. However, the comparison would be clearer if the authors provided more details about the inference-time setup of the baselines, especially HardNet-Aff. Overall, the evidence supports the main constraint-satisfaction claim, but the results also reveal important trade-offs in task accuracy, computational cost, and the robustness of the heuristic sign-assignment procedure.

**Requested Changes:**

### Major comments

1. As mentioned in the manuscript, since the activation cell is typically larger than the target region, local flexibility is sacrificed, which leads to a larger loss. I wonder whether this issue can be mitigated by selecting a proper value of (\epsilon) in Figure 2, or whether additional constraints are needed to force the vertices of the target region and the activation cell to be better aligned. Is it also possible to quantify the trade-off between the loss and the size of the activation cell?

2. Although Table 3 provides a comparison of the running times of different affinity-enforcement methods, it is not clear to me how (\epsilon) is selected. If the same value of (\epsilon) is applied to all affinity-enforcement methods, one might expect these three approaches to achieve the same level of constraint satisfaction. However, in Table 5 of the appendix, ADMM achieves the best violation and MSE. More discussion regarding the geometric properties of these different affinity-enforcement methods would be beneficial. I also wonder whether a mixture of different affinity-enforcement methods could lead to better overall performance in practice.

3. In Figure 5, it seems that the dependence of computation time on the number of constrained vertices is very similar for the three methods, which is somewhat counterintuitive. For example, the computational cost of ADMM might be expected to grow faster as the number of vertices increases.

4. The sign-assignment procedure appears to be heuristic to some extent. I wonder whether there are cases where the affinity-enforcement problem becomes infeasible for a specific sign assignment; for example, two target regions may not be enforceable as affine at the same time under that sign assignment. If a sign assignment fails, is there a way to detect and fix it?

5. To increase local flexibility, one intuitive idea is to divide one target region into several sub-regions and assign each sub-region an individual affine pattern. How technically practical is this idea?

6. I have one clarification question regarding the benchmark in Table 1. The inference time of HardNet-Aff is much larger than that of the other methods. This is difficult to interpret without knowing whether HardNet-Aff uses the same base network architecture, or whether it requires an additional inference-time projection or correction step. Clarifying these details would make the empirical comparison more transparent.

### Minor comments

1. The definition of “Affine RMSE” is not sufficiently clear. Since this metric is central to the claim that mPOLICE enforces region-wise affinity while HardNet-Aff only enforces point-wise feasibility, the paper should provide an explicit mathematical definition.

---

> ### Author Response · Authors · 2026-07-15
> **Reviewer vkqH**
>
> Thank you for the careful, constructive questions. We added controlled Experiments E9–E13 in Appendix B. E10 studies the loss and activation-cell-size trade-off, E9 compares enforcement backends and their scaling, and E12 tests infeasible sign assignments and repair. We also distinguish `tau` for assignment, `delta` for the sign margin, `eta` for constraint tightening, and `epsilon` for optional early stopping. Algorithm 1 and Tables 10–11 report the configurations and verification modes.
>
> **1. Can epsilon mitigate the lost flexibility, and can the loss/cell-size trade-off be quantified?** No. `epsilon` is an optional task-scale stopping heuristic and does not control activation-cell size. Exact alignment would require relevant ReLU boundaries to match the region facets, which mPOLICE does not impose. To quantify the trade-off, we added E10. It shows that larger sign margins can enlarge the containing cell and hurt fit, while added width can reduce the observed gap. Neither trend is a theorem or a guaranteed tuning rule.
>
> **2. Why don't all backends reach the same satisfaction level, and would a hybrid help?** Under matched settings, every backend that passes the common gate reaches the same satisfaction status. Differences come from stopping and the feasible point returned, not different constraint sets. To test this and the hybrid directly, we added E9. It fixes the checkpoint, sign assignment, margin, and gate while comparing QP, POCS, ADMM, and POCS-to-QP. All matched runs pass with zero measured dense violation. QP gives the minimum-change solution per neuron to solver tolerance, POCS need not return the nearest point, and capped ADMM can miss the requested margin. The hybrid is a viable polishing strategy, but E9 shows no clear timing or fit advantage.
>
> **3. Why did the timing curves look so similar across methods?** The curves looked similar because complete calls over only one to four regions include common workflow costs and variable solver iterations that obscure backend differences. To address this, we added two complementary views. Figure A1 now reports complete verified calls, and Section 4.5 and E9 add an isolated vertex-count sweep. The isolated sweep reveals the expected separation: POCS and capped ADMM stay in the millisecond range over the tested vertices, while per-neuron QP grows sharply. We report both as measurements, not fitted complexity laws. Appendix P gives the asymptotic accounting.
>
> **4. Can a sign assignment be infeasible, and can that be detected and fixed?** Yes. To demonstrate detection and repair, we added E12. It constructs a `(+,-,+)` assignment across three ordered collinear regions that no first-layer affine pre-activation can realize. The solver reports infeasibility and the signed-margin check localizes the failing neuron. A bounded heuristic flips that neuron's least-committed region, rechecks uniqueness, and succeeds after re-enforcement. This is one successful case. A failed margin check is not a general infeasibility proof, and repair is not complete.
>
> **5. Is sub-region splitting practical?** It can help at modest subdivision levels, but it is not guaranteed to remain feasible as refinement increases. To test this, we added E7 in Appendix A. Initial refinements lower interpolation error, followed by configuration-specific failures for finer partitions. Distinct positive-margin pieces require gaps outside the guarantee, and more pieces increase the vertex and assignment burden. We therefore make no monotonic convergence claim.
>
> **6. Why is HardNet-Aff's inference time so much larger?** HardNet-Aff is slower because it applies a closed-form correction on every forward pass, whereas mPOLICE leaves the deployed graph unchanged. We clarified Section 4.1 by stating that every arm starts from the same pretrained network and by identifying this inference-time correction. HardNet-Aff performs well for the tested pointwise output inequalities. mPOLICE instead establishes one affine map on each target region, enabling local-map and first-derivative constraints. We do not present this as a task-accuracy advantage.
>
> **7. Affine RMSE.** We added its explicit mathematical definition in Equation (12). It is the residual of the least-squares best affine fit on dense in-region samples. It diagnoses sampled affinity but does not certify it. The whole-region conclusion comes from verified sign and vertex conditions and Theorems 1–2, not from a small sampled RMSE.